# Metavinculin modulates force transduction in cell adhesion sites

Verena Kanoldt[1,2], Carleen Kluger[2], Christiane Barz[2], Anna-Lena Schweizer [1,2], Deepak Ramanujam[3,4], Lukas Windgasse[1], Stefan Engelhardt [3,4], Anna Chrostek-Grashoff[1,2] & Carsten Grashoff [1,2]✉

Vinculin is a ubiquitously expressed protein, crucial for the regulation of force transduction in cells. Muscle cells express a vinculin splice-isoform called metavinculin, which has been associated with cardiomyopathies. However, the molecular function of metavinculin has remained unclear and its role for heart muscle disorders undefined. Here, we have employed a set of piconewton-sensitive tension sensors to probe metavinculin mechanics in cells. Our experiments reveal that metavinculin bears higher molecular forces but is less frequently engaged as compared to vinculin, leading to altered force propagation in cell adhesions. In addition, we have generated knockout mice to investigate the consequences of metavinculin loss in vivo. Unexpectedly, these animals display an unaltered tissue response in a cardiac hypertrophy model. Together, the data reveal that the transduction of cell adhesion forces is modulated by expression of metavinculin, yet its role for heart muscle function seems more subtle than previously thought.

---

[1] Department of Quantitative Cell Biology, Institute of Molecular Cell Biology, University of Münster, 48149 Münster, Germany. [2] Max Planck Institute of Biochemistry, Group of Molecular Mechanotransduction, 82152 Martinsried, Germany. [3] Institute of Pharmacology and Toxicology, Technical University of Munich, 80802 Munich, Germany. [4] DZHK (German Centre for Cardiovascular Research), Partner Site Munich Heart Alliance, 80802 Munich, Germany. ✉email: grashoff@wwu.de

The ability of cells to sense and respond to mechanical stress is crucial for many developmental and postnatal homeostatic processes, and especially critical in tissues naturally exposed to significant mechanical loads[1,2]. Mechanical signals between cells and the extracellular matrix (ECM) are processed in macromolecular structures called focal adhesions (FAs), which assemble around integrin receptors and mediate the connection to the actin cytoskeleton[1–3]. Central to the mechanosensitivity of FAs is vinculin, a ubiquitously expressed actin-binding protein that is recruited to FAs upon force-induced association with the integrin activator talin. As a connector between talin and the actin cytoskeleton, vinculin strengthens FAs and modulates force transmission[4–6]. Vinculin is thought to play a similar role in adherens junctions (AJs), to which it is recruited upon force-sensitive binding to α-catenin[7,8].

Intriguingly, some mammalian tissues, in particular muscle cells, express a vinculin splice variant called metavinculin[9–11]. This isoform differs from vinculin by a 68-amino-acid (aa)-long insert in the C-terminal region of the molecule[12,13] leading to distinct actin filament organization[14–16]. As the interaction with actomyosin is essential for the ability of vinculin to transduce mechanical signals[5], it has been speculated that intracellular force propagation may be vinculin isoform-specific[15,17]. However, the mechanical role of metavinculin remained obscure because sufficiently sensitive technologies to evaluate the molecular mechanics of metavinculin in living cells were still missing. In this study, we have applied a set of piconewton (pN)-sensitive tension sensors (TSs)[5,18,19] to explore the mechanics of vinculin and metavinculin in cell adhesions using live-cell fluorescence lifetime imaging microscopy (FLIM).

The loss of the metavinculin isoform[20] and mutations in metavinculin[21,22] were identified in patients suffering from dilated and hypertrophic cardiomyopathies, thus the role of metavinculin is often discussed in the context of heart muscle disorders. However, family linkage analyses[23] to establish genetic causality between metavinculin-mutations and cardiomyopathies are missing, and conclusive experimental evidence for metavinculin dysfunction causing heart muscle disorders is absent. Therefore, we have also generated metavinculin-deficient mice to evaluate the relevance of metavinculin loss for heart muscle pathophysiology in vivo.

Together, our cell culture experiments show that metavinculin expression modulates how molecular forces are transduced in cell adhesion sites, while the evaluation of knockout mice reveals that the role of metavinculin for heart muscle function is not as critical as previously thought.

## Results

**Enhanced talin association leads to immobilization of metavinculin in FAs.** To systematically investigate the role of metavinculin, we generated venus-tagged vinculin (V-V) and metavinculin (M-V) constructs (Supplementary Fig. 1a) and expressed them in vinculin-deficient mouse embryonic fibroblasts (vinc$^{(-/-)}$)[18,24]. Both proteins independently localized to FAs and rescued the spreading defect of vinc$^{(-/-)}$ cells, which display a significantly reduced cellular eccentricity 2 h after seeding, compared to the control cell line (Fig. 1a, b). The cell and FA morphology, but also the organization of the actin cytoskeleton and the expression of central FA proteins, were indistinguishable between vinc$^{(-/-)}$ cells reconstituted with either V-V or M-V (Supplementary Fig. 1b, c). Moreover, co-expression of both isoforms revealed a virtually complete overlap of both proteins in cell adhesion sites (Fig. 1c).

We next seeded V-V- and M-V-expressing cells on micropatterned surfaces, upon which cells form FAs of uniform size

and intensity, to investigate the subcellular dynamics by fluorescence recovery after photobleaching (FRAP) experiments (Supplementary Fig. 2a, b). Again, the FA morphologies of both cell lines were indistinguishable, and fluorescence recovery rates of co-expressed TagBFP-HA-tagged talin-1 (T1-B-HA; Supplementary Fig. 1a) were similar indicating that the overall FA dynamics are comparable (Fig. 1d). Consistent with a previous report[25], however, the mobile fraction of metavinculin was significantly lower when compared with vinculin (Fig. 1e). As the primary binding partner of vinculin in FAs is talin, we tested whether an altered talin association may underlie the reduced mobility of metavinculin and, indeed, talin was enriched in M-V immunoprecipitates (Fig. 1f). To validate this finding, we co-expressed T1-B-HA in V-V- and M-V-expressing cells and performed HA-mediated immunoprecipitations. As expected, M-V was significantly enriched in the talin-1 pulldown (Fig. 1g). Together, these experiments demonstrate that metavinculin can compensate for the loss of vinculin with regard to FA formation and cell spreading, but a larger fraction of metavinculin is immobilized in FAs presumably because of enhanced talin-binding.

**Force transduction in FAs is vinculin isoform-dependent.** We previously showed that vinculin is exposed to pN-scale forces in FAs, where it modulates force transduction across the integrin–talin linkage[5,18,19]. To investigate whether vinculin and metavinculin propagate mechanical forces differently, we generated vinculin- (V-TS) and metavinculin-based (M-TS) TSs using four single-molecule-calibrated modules sensitive to 1–6 pN (F40)[5], 3–5 pN (FL)[19], 6–8 pN (HP35)[18], and 9–11 pN (HP35st)[18] that were inserted between the (meta)vinculin head and tail domain, after aa 883. In parallel, we generated control constructs to determine the fluorescence lifetime of the donor fluorophore as well as the FRET efficiency of the no-force control (Con-TS), which comprises the vinculin head domain (aa 1–883) and a TS module but lacks the vinculin tail domain (Supplementary Fig. 3a–c). V-TS and M-TS localized to FAs in vinc$^{(-/-)}$ cells (Fig. 2a and Supplementary Fig. 3d) and rescued their spreading phenotype equally (Fig. 2b and Supplementary Fig. 4), confirming the initial observation that metavinculin can compensate for vinculin loss in this cell type. The Con-TS also localized to adhesion sites but induced slightly hypertrophic FAs, as reported earlier[4,5] (Fig. 2a). Furthermore, actin co-sedimentation assays[5,26] with lysates from HEK293 cells expressing V-V, V-TS, M-V, and M-TS in the presence or absence of the vinculin activator IpaA confirmed that TS module insertion does not lead to constitutive activation of the vinculin isoforms (Supplementary Fig. 5).

Since our previous studies on talin mechanics revealed molecular forces as high as 11 pN[18], we started our tension measurements using HP35st-based constructs that respond to such force magnitudes. Using our previously published data analysis workflow[18,19], illustrated in Supplementary Fig. 6, we detected a marked decrease in FRET efficiency in V-TS- and M-TS-expressing cells, indicating that mechanical forces of at least 9–11 pN occur across (meta)vinculin junctions in FAs (Supplementary Fig. 7a). We note that these data are consistent with a recently published single-molecule force spectroscopy study providing in vitro evidence for the presence of comparably high forces across talin and vinculin linkages[8].

Intriguingly, the FRET efficiency decrease was consistently less pronounced in M-TS expressing cells (Supplementary Fig. 7a). To control for the specificity of this observation, we inserted a previously reported I997A mutation into vinculin (V-TS-I997A)[24,27] and—at the corresponding residue—into metavinculin (M-TS-I1065A), reducing actin engagement and

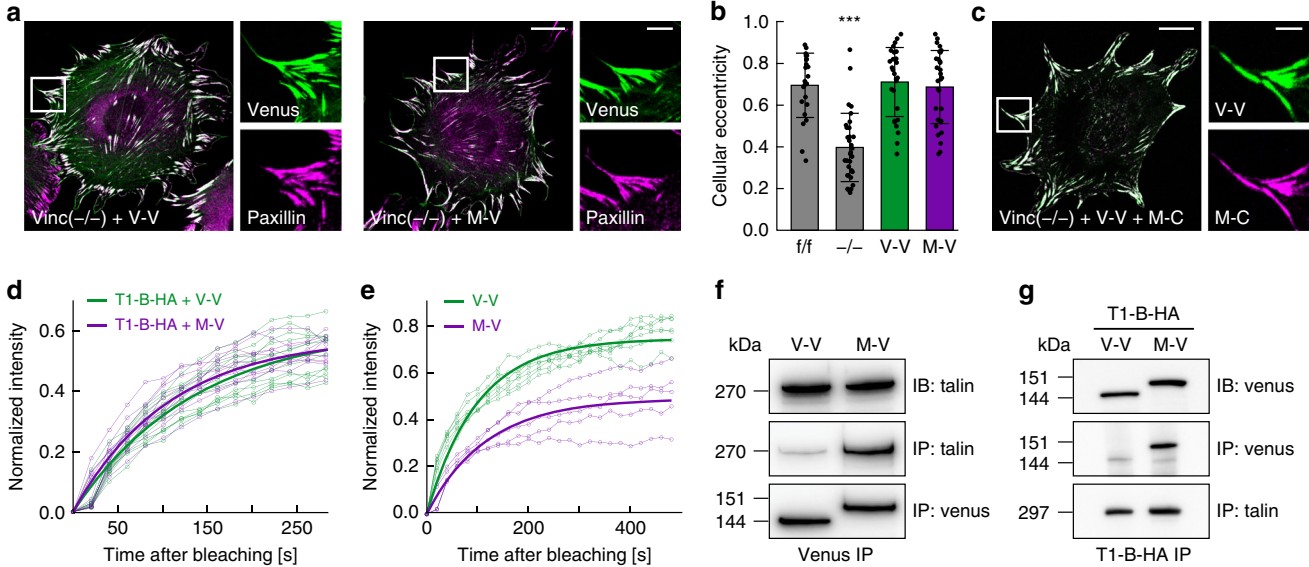

**Fig. 1 Metavinculin is immobilized in focal adhesions (FAs) and displays enhanced binding to talin. a** Representative images of vinculin-deficient (vinc$^{(-/-)}$) cells expressing vinculin–venus (V-V) or metavinculin–venus (M-V). Co-staining with a paxillin antibody indicates the independent localization of each vinculin isoform to FAs. **b** V-V and M-V both rescue the spreading defect of vinc$^{(-/-)}$ cells, which display reduced cellular eccentricity 2 h after cell adhesion when compared to the parental cell line (vinc$^{(f/f)}$). ($n = 23, 32, 27, 30$ cells). Two-sided Kolmogorov–Smirnov test: ***$p < 0.001$. The bar chart shows the mean values ± SD. **c** V-V and metavinculin–mCherry (M-C) co-localize in FAs when co-expressed in vinc$^{(-/-)}$ cells. **d** Talin turnover rates are comparable in V-V- or M-V-expressing vinc$^{(-/-)}$ cells, as revealed by FRAP analysis of cells with co-expressed talin-1-TagBFP-HA (T1-B-HA). ($n = 12, 12$ cells). **e** Metavinculin resides in FA longer than vinculin as shown by FRAP analysis of vinc$^{(-/-)}$ cells expressing V-V or M-V. ($n = 7, 5$ cells).
**f** Co-immunoprecipitation (IP) experiments using the venus-tag as bait demonstrate an increased association of talin with metavinculin (IB: immunoblot).
**g** Metavinculin is enriched in an HA-tag-driven IP performed on cells co-expressing T1-B-HA and V-V or M-V. Scale bars indicate 20 μm, in zoom: 5 μm. Expected molecular weight values are indicated (kDa). Source data, exact $p$ values, and uncropped immunoblots with protein markers are provided in the Source Data file.

thus molecular forces experienced by both vinculin isoforms. Indeed, these mutations increased the FRET efficiencies to almost no-force control levels and eliminated vinculin isoform-specific differences (Supplementary Fig. 7a, e). We confirmed both effects with an independent set of measurements using analogous FL-based (meta)vinculin force sensors (Fig. 2c) and excluded unspecific effects through intermolecular FRET[28,29], which was comparably small and even slightly lower in metavinculin-expressing FAs (Supplementary Fig. 7b, c). To further validate that the FRET difference between vinculin and metavinculin is force-dependent, we expressed FL-based V-TS, M-TS, and Con-TS in talin-deficient cells (tln1$^{-/-}$tln2$^{-/-}$), which do not form FAs and in which all constructs are localized in the cytoplasm[30]. We seeded these cells onto poly-L-Lysine (pLL)-coated dishes and treated them with Y-27632 inhibitor to ensure the absence of mechanical forces. As expected, FRET–FLIM measurements of these cells did not reveal any difference between V-TS, M-TS, and Con-TS FRET efficiencies (Fig. 2d). Finally, we replaced the mechanosensitive linker in the TS module with a flexible 7-aa-long peptide (F7) that cannot be significantly elongated under force. The resulting force-insensitive vinculin and metavinculin controls, V-F7-TS and M-F7-TS, expressed in vinc$^{(-/-)}$ cells displayed highly similar FRET efficiencies showing that the isoform-specific effect is not caused by different conformations of the vinculin isoforms (Fig. 2e). Together, the experiments demonstrate that the observed effects are specific, and force transduction in FAs is vinculin isoform-dependent.

**Metavinculin expression modulates force transduction across the integrin–talin linkage**. To investigate whether the differences

in molecular forces across vinculin and metavinculin affect force transduction across the integrin-talin linkage, we reconstituted vinculin-deficient cells with TagBFP-tagged vinculin (V-B) or metavinculin (M-B) and co-expressed a talin-2 TS (T2-TS)[18], which harbors the HP35 TS module between the head and rod domain, after aa 450 (Supplementary Fig. 7d); the no-force control comprises the TS module at the C-terminus of talin-2 (T2-Con). We focused our experiments on this particular talin-isoform, as the expression of metavinculin and talin-2 correlate in muscle tissues[31]. Consistent with our earlier study[18], forces acting across talin-2 were small in the absence of vinculin expression, whereas the presence of vinculin strongly elevated talin-2 forces. By contrast, expression of metavinculin only slightly increased talin-2 tension (Fig. 2f) showing that force propagation across the integrin–talin junction is vinculin isoform-dependent.

Remarkably, the data indicated that metavinculin transduces mechanical forces ostensibly less efficiently than vinculin, even though it displayed an enhanced interaction with talin (Fig. 1f, g). This suggested that the association of (meta)vinculin with talin and force transduction are separate events that do not necessarily correlate. To test this hypothesis, we inserted an A50I mutation—which reduces vinculin binding to talin[32,33]—into the FL-based TSs (V-TS-A50I and M-TS-A50I) (Supplementary Fig. 7e) and expressed these constructs in vinc$^{(-/-)}$ cells. FLIM–FRET analysis revealed that the A50I mutation lowered metavinculin FRET efficiencies to levels observed in V-TS cells, while even further reducing them in vinculin samples (Fig. 2g). Since a previous study reported a shift towards slightly increased FRET values in A50I mutants[34], we repeated these experiments with analogous F40-based TS constructs. However, consistent with the FL-based measurements, we observed again a FRET efficiency decrease in all A50I mutants (Fig. 2h). Our data, therefore,

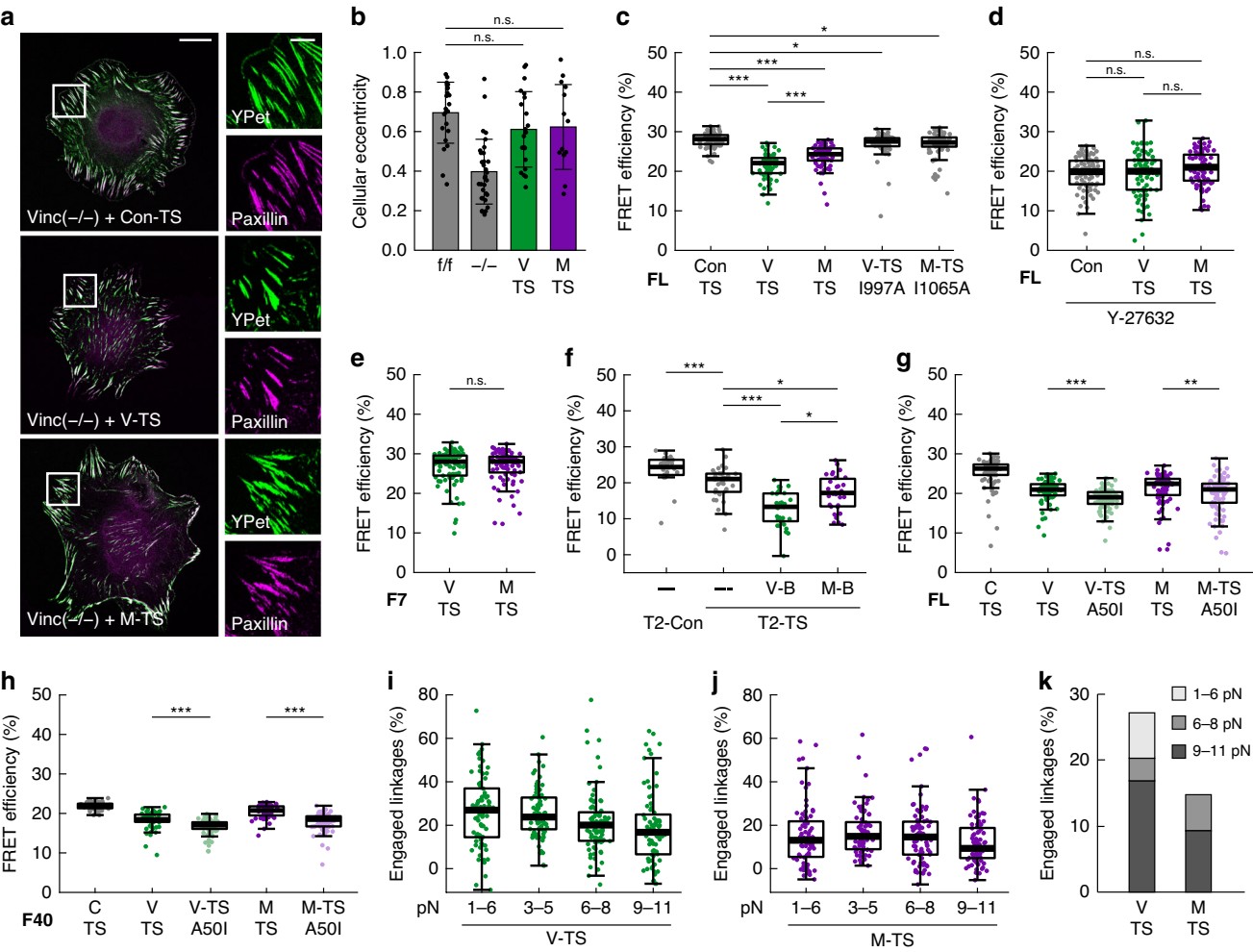

**Fig. 2 Force transduction in FAs is vinculin isoform-dependent. a** Representative images of vinculin-deficient (vinc$^{(-/-)}$) cells expressing vinculin tension sensor (V-TS), metavinculin tension sensor (M-TS), and the no-force control (Con-TS) 4 h after spreading on FN-coated glass coverslips show localization of all constructs to FAs (YPet), which are visualized by paxillin staining. Scale bar: 20 µm, in zoom: 5 µm. **b** Expression of V-TS or M-TS rescues the spreading defect of vinc$^{(-/-)}$ cells; data of the parental (vinc$^{(f/f)}$) and vinc$^{(-/-)}$ cells are the same as in Fig. 1b. ($n = 23, 32, 21, 14$ cells). The bar chart shows the mean values ± SD. **c** Live-cell FLIM measurements of vinc$^{(-/-)}$ cells expressing FL-based tension sensors demonstrate FRET efficiency differences between V-TS and M-TS. Impairing actin binding by inserting the I997A mutation into vinculin (V-TS-I997A) and I1065A into metavinculin (M-TS-I1065A) strongly reduces tension and eliminates vinculin isoform-specific differences. ($n = 73, 73, 74, 72, 73$ cells). **d** Live-cell FLIM measurements of FL-based Con-TS, V-TS, and M-TS expressed in talin-deficient cells (tln1$^{-/-}$tln2$^{-/-}$), seeded on pLL-coated dishes and treated with Y-27632, confirmed that FRET differences are force-specific. ($n = 80, 80, 82$ cells). **e** Highly similar FRET efficiencies of force-insensitive vinculin (V-F7-TS) and metavinculin (M-F7-TS) tension sensor controls expressed in vinc$^{(-/-)}$ cells demonstrate that vinculin isoform-specific effects are conformation-independent ($n = 86$, 85 cells). **f** Talin-2 tension sensor (T2-TS) measurements in vinc$^{(-/-)}$ cells expressing TagBFP-tagged vinculin (V-B) and metavinculin (M-B) show that vinculin isoform-specific force transduction propagates across talin-2. T2-Con: talin-2 no-force control. ($n = 30, 36, 31, 31$ cells). **g, h** The A50I point mutation, which reduces the binding affinity of (meta)vinculin to talin, caused a FRET efficiency decrease in FL- and F40-based vinculin (V-TS-A50I) and metavinculin (M-TS-A50I) samples. (**g**: $n = 84, 83, 78, 85, 86$ cells; **h**: $n = 60, 59, 77, 57, 78$ cells). **i** Examination of stretched sensor molecules in V-TS-expressing cells, using four different TS modules, shows that vinculin is exposed to a wide range of forces; in average, 20–30% of molecules experience mechanical tension ($n = 77, 73, 81, 77$ cells). **j** Analogous analysis of M-TS-expressing cells indicates that the fraction of mechanically engaged metavinculin molecules is <20%. Note the equal amounts of stretched molecules in samples containing sensors sensitive to 1–6 pN, 3–5 pN, and 6–8 pN indicating comparably high force per molecule across metavinculin. ($n = 80, 74, 80, 77$ cells). **k** Analyzing the differences of medians shown in (**i**) and (**j**) indicates that cells expressing M-TS instead of V-TS have less mechanically-engaged linkages that experience higher tension per molecule. Boxplots show median, 25th and 75th percentile with whiskers reaching to the last data point within 1.5× interquartile range. Two-sided Kolmogorov–Smirnov test: ***$p < 0.001$, **$p < 0.01$, *$p < 0.05$, n.s. (not significant) $p \geq 0.05$. Source data and exact $p$ values are provided in the Source Data file.

suggest that a decrease (or increase) in talin association does not directly translate into a decrease (or increase) of (meta)vinculin force transduction. We note that this result is not only consistent with single-molecule force spectroscopy studies showing that vinculin binding to talin requires talin tension but no mechanical forces across vinculin[35,36]; it is also in line with a recent study demonstrating that a force-independent relief of talin/vinculin

autoinhibition is sufficient to mediate a tight interaction between both proteins[37].

**Metavinculin displays a lower engagement ratio but higher force per molecule and also modulates force transduction in cell–cell junctions.** The results above seemed to suggest that

metavinculin experiences, at least on average, lower forces per molecule than vinculin. Alternatively, a smaller FRET efficiency reduction could be caused by a lower fraction of mechanically engaged molecules[19,38]. To distinguish these two scenarios, we generated and analyzed cell lines expressing F40-, FL-, HP35-, and HP35st-based vinculin and metavinculin TSs and determined the fraction of stretched molecules at different force levels using our previously published bi-exponential fitting algorithm[19] (Supplementary Fig. 6c). For vinculin-expressing cells, the fraction of engaged molecules decreased with increasing sensor stiffness indicating that vinculin is exposed to a range of forces between 1 and 11 pN (Fig. 2i and Supplementary Fig. 8a, b). By contrast, engagement ratios in metavinculin-expressing cells were lower but remained rather constant over a wider force range and only decreased at forces larger than 8 pN (Fig. 2j). In conclusion, these data reveal that a smaller fraction of metavinculin junctions bear mechanical forces; the linkages that are engaged, however, carry higher mechanical loads (Fig. 2k).

Metavinculin is typically expressed in smooth-muscle and striated-muscle cells as well as cardiomyocytes; but not in fibroblasts[10,11,39,40]. Because of the proposed function of metavinculin in the heart muscle, we investigated whether the isoform-specific effects are also observed in HL-1 cells, a model cell line derived from mouse atrial cardiomyocytes[41] that naturally expresses metavinculin (Supplementary Fig. 9). Expression of FL-based V-TS and M-TS as well as the no-force control (Con-TS) in HL-1 cells revealed the expected localization to FAs but also to cell–cell junctions when cells were cultured at sufficiently high densities (Fig. 3a, c). Again, we did not observe differences in subcellular localization between the vinculin isoforms, and FRET efficiencies indicated mechanical tension across them both, albeit forces in FAs seemed smaller in this cell type. Consistent with our data in fibroblasts, FRET efficiencies were lower and spread over a wider range in vinculin-expressing

cells (Fig. 3b). Intriguingly, we observed this effect also in cell–cell junctions of HL-1 cells suggesting that metavinculin is able to tune force transduction in AJs as well (Fig. 3d). Thus, vinculin isoform-specific force transduction is a conserved phenomenon, found in different cell types and cell adhesion complexes.

**Metavinculin-deficient mice display an unaltered tissue architecture and a normal hypertrophic response.** The absence or mutation of metavinculin were observed in cardiomyopathy patients, thus metavinculin is widely considered a cardiomyopathy gene[20–22,42,43]. Yet the number of identified patients is still comparably small and direct evidence for the causative role of metavinculin dysfunction for cardiomyopathies, for example in form of family linkage analysis, is missing. We, therefore, decided to evaluate the importance of metavinculin expression for heart muscle function in mice by generating metavinculin knockout animals ($M^{(-/-)}$) (Supplementary Fig. 10a). PCR analysis confirmed the excision of the targeted exon (Supplementary Fig. 10b), while Western blot analysis established a reduction of metavinculin expression in heterozygous animals and a complete loss of metavinculin in homozygotes (Fig. 4a). In control animals, as shown before[44], metavinculin was expressed at high levels in the uterus and moderate levels in heart and skeletal muscle tissues. $M^{(-/-)}$ mice are born at the expected mendelian ratio (Supplementary Fig. 10c), they do not display an overt phenotype, age normally, and are fertile. Histological analysis of heart muscle sections revealed an intact tissue architecture (Fig. 4b), while costamere, intercalated disc (ICD), and gap junction (GJ) proteins were normally expressed and localized in 6- and 13-month-old animals (Fig. 4c, d and Supplementary Figs. 11 and 12).

To explore the importance of metavinculin expression under pathological conditions, we exposed 8-week-old $M^{(-/-)}$ and wild-type ($M^{(+/+)}$) mice to a transverse aortic constriction (TAC)

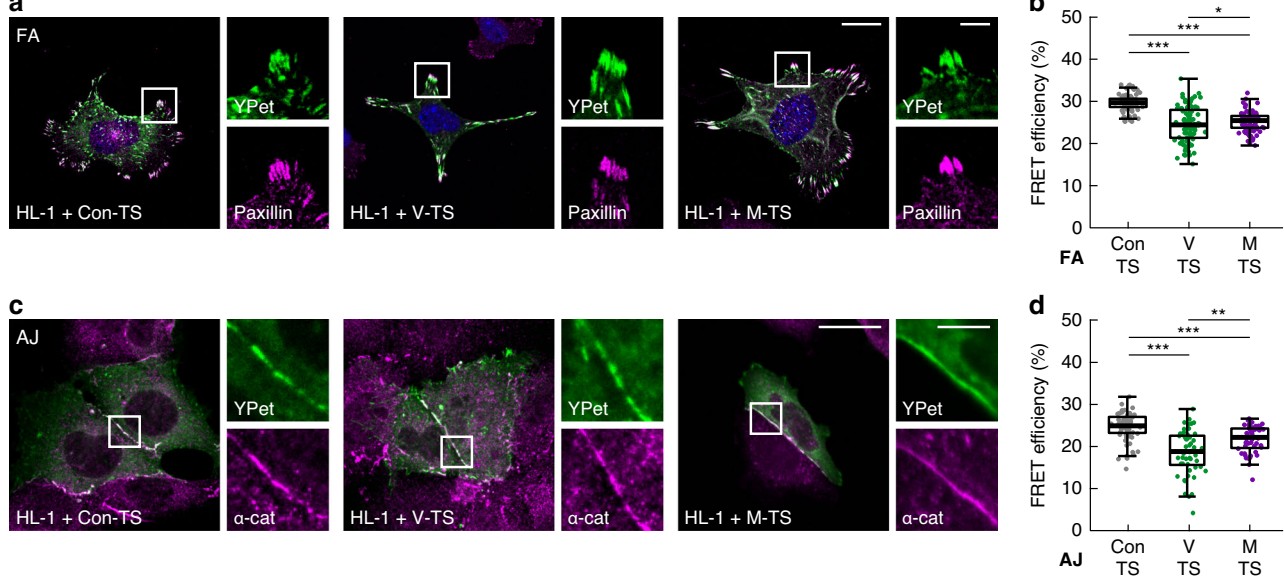

**Fig. 3 Vinculin isoform-specific differences in force transduction are observed in FAs and adherens junctions (AJs) of muscle cells. a** Representative images of HL-1 cells expressing FL-based vinculin tension sensor (V-TS), metavinculin tension sensor (M-TS), and the no-force control (Con-TS) show localization of all fusion proteins to FAs (YPet), which are co-stained with a paxillin antibody. Nuclei are visualized with DAPI (blue). **b** FLIM measurements of FAs in HL-1 cells expressing FL-based sensors demonstrate vinculin isoform-specific force transduction also in muscle cells. ($n = 81, 92, 84$ cells). **c** Representative images of HL-1 cells expressing V-TS, M-TS, and Con-TS show localization of all tension sensors to AJs (YPet). The signal of co-stained α-catenin (α-cat) is used to outline AJs. **d** FLIM measurements of AJs in HL-1 cells reveal vinculin isoform-specific force transduction in cell–cell contacts. ($n = 65, 53, 48$ cells). Boxplots show median, 25th, and 75th percentile with whiskers reaching the last data point within 1.5× interquartile range. Two-sided Kolmogorov–Smirnov test: ***$p < 0.001$, **$p < 0.01$, *$p < 0.05$. Scale bar: 20 μm, in zoom: 5 μm. Source data and exact $p$ values are provided in the Source Data file.

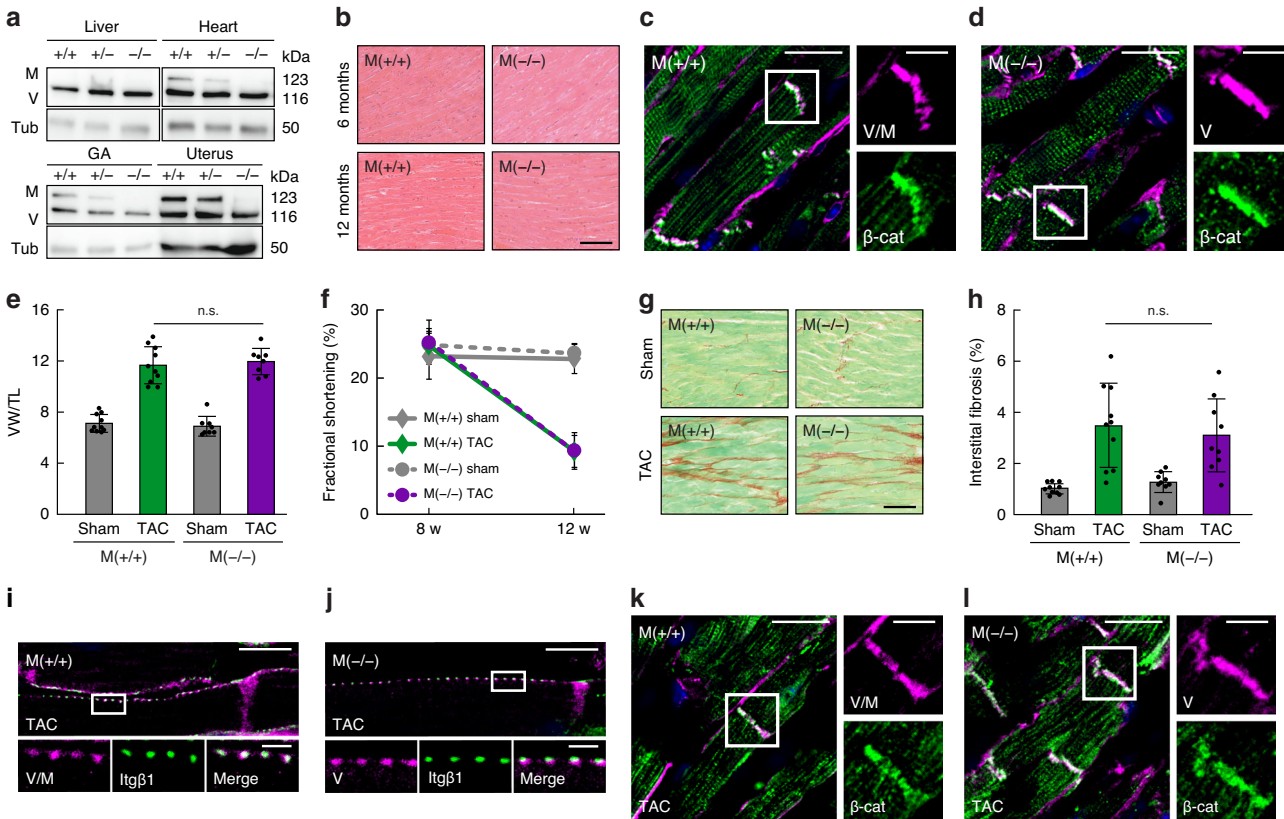

**Fig. 4 Metavinculin knockout mice display a normal hypertrophic response upon transverse aortic constriction (TAC). a** Western blot analysis of tissue lysates from 6-month-old mice shows complete loss of metavinculin in knockout M$^{(-/-)}$ and ~50% reduction of expression in heterozygous M$^{(+/-)}$ animals, compared to wild-type M$^{(+/+)}$ littermates. (GA: gastrocnemius muscle, V: vinculin, M: metavinculin, Tub: tubulin). **b** Histological analysis of M$^{(+/+)}$ and M$^{(-/-)}$ heart tissue of 6- and 13-month-old mice indicates normal morphology of the cardiac muscle. Scale bar: 40 μm. **c, d** Representative immunohistochemistry of ventricular tissue from 13-month-old M$^{(+/+)}$ and M$^{(-/-)}$ mice, respectively, reveals unchanged intercalated disk (ICD) structures as indicated by β-catenin (β-cat) and (meta)vinculin (V/M) staining. Nuclei are visualized with DAPI (blue). Scale bar: 20 μm, in zoom: 10 μm. **e** Heart weight analysis of sham- or TAC-operated mice, expressed as the ventricular weight (VW) normalized to tibia length (TL) shows no difference between M$^{(+/+)}$ and M$^{(-/-)}$ mice ($n = 10, 10, 9, 9$ mice). The bar chart shows the mean values ± SD. **f** Echocardiography assessment shows the expected decrease in fractional shortening in TAC-operated animals; the effect is highly similar in M$^{(+/+)}$ and M$^{(-/-)}$ mice. ($n = 10, 9, 9, 7$ mice). The line chart shows mean values ± SD. **g** Representative images of Sirius Red/Fast Green-stained myocardium show anticipated tissue fibrosis in each group of TAC-operated animals. Scale bar: 40 μm. **h** Quantitative analysis of interstitial fibrosis reveals a normal tissue response in metavinculin-deficient mice. ($n = 10, 10, 9, 9$ mice). **i, j** Representative immunohistochemistry of ventricular tissues from 3-month-old M$^{(+/+)}$ and M$^{(-/-)}$ TAC-operated mice, respectively. Costameres are visualized with β1 integrin (Itgβ1) and (meta)vinculin (V/M) staining. Scale bar: 10 μm, in zoom: 2 μm. **k, l** Representative immunohistochemistry of ventricular tissues from 3-month-old M$^{(+/+)}$ and M$^{(-/-)}$ TAC-operated mice shows normal ICD structures as indicated by β-catenin (β-cat) and (meta)vinculin (V/M) staining. Nuclei are visualized with DAPI (blue). Scale bar: 20 μm, in zoom: 10 μm. Two-sided ANOVA followed by Sidak's test: n.s. (not significant) $p \geq 0.05$. Expected molecular weight values are indicated (kDa). Source data, exact $p$ values, and uncropped immunoblots with protein markers are provided in the Source Data file.

protocol (Supplementary Fig. 13a), an experimental model for pressure overload-induced cardiac hypertrophy[45]. As expected, control M$^{(+/+)}$ animals exposed to TAC showed an increase in ventricle weight 4 weeks after TAC, when compared to sham-operated animals (Fig. 4e and Supplementary Fig. 13b). Echo-cardiography revealed a decreased fractional shortening (Fig. 4f and Supplementary Table 1), histology of the myocardium indicated tissue fibrosis upon pressure overload (Fig. 4g, h), and RT-qPCR analysis confirmed the expected changes in marker gene expression (Supplementary Fig. 13c). Remarkably, M$^{(-/-)}$ animals displayed a hypertrophic response upon TAC that was indistinguishable from the control group (Fig. 4e–h). The overall structures of costameres, ICDs, and GJs were also comparable between both cohorts as indicated by the localization of respective marker proteins such as β1 integrin, β-catenin, and connexin-43 (Fig. 4i–l and Supplementary Fig. 14). We, therefore, conclude that the loss of metavinculin does not impair development and homeostasis of the analyzed muscle tissues and does not cause, under conditions applied in this study, an aberrant hypertrophic response in mice.

## Discussion

For a long time, since the discovery of metavinculin almost 40 years ago[9], the function of this vinculin splice-isoform remained elusive. A number of excellent biochemical[14,46] and structural[13,16] analyses revealed that the presence of the meta-vinculin insert in the C-terminal domain of vinculin leads to a distinct association with and bundling of actin filaments. Given the established role of vinculin as a force transducer[4,5,18,47,48], it seemed intuitively obvious that metavinculin expression may somehow modulate force propagation in cells, and cell culture studies indeed indicated that a range of cellular processes can be differentially regulated in a vinculin isoform-dependent

fashion[17,25]. Yet, direct evidence for metavinculin regulating molecular force transduction in cells was missing.

In this study, we show that metavinculin expression leads to an alternate mode of force propagation in cell adhesion sites. Metavinculin displays an increased association with talin, which is consistent with the recently published observation of metavinculin being partially activated[49], leading to a decreased turnover rate in FAs[25]. Intriguingly, the increased talin association does not translate into an increased force propagation across metavinculin, because the fraction of metavinculin molecules experiencing mechanical tension is smaller as compared to vinculin. Those metavinculin junctions that are under tension, however, bear higher mechanical loads. Together, this indicates that metavinculin acts as a modulator of cell adhesion mechanics, even though the functional consequences of metavinculin-mediated force transduction remain to be defined in more detail. A possible role, consistent with our data, is that metavinculin serves as a mechanical buffer protein: If mechanical forces are born by a smaller fraction of linkages, the increased number of unloaded molecules may allow resisting a sudden force increase across the cell adhesion structure.

Besides these biological implications, our results emphasize two important technical aspects: First, it appears to be critical to apply a range of TS modules in parallel rather than relying on individual probes with inherently limited force sensitivity. Second, it is crucial to determine, in addition to the average force value, the amount of mechanically engaged molecules[19,38]. Both parameters together—the average force per molecule (increased for metavinculin) and the fraction of mechanically engaged proteins (decreased for metavinculin)—define how molecular linkages transduce mechanical signals.

The obvious question, however, is how the distinct mechanical properties of metavinculin could be linked to a potential role as a cardiomyopathy gene. It has been demonstrated in mice that reducing the expression levels of both vinculin isoforms predisposes rodents to developing cardiomyopathies[50], while specifically removing vinculin, and thus metavinculin, from cardiac muscle cells induces cardiomyopathy within 6 months[51]. However, conclusions on isoform-specific effects are difficult to draw from these studies because metavinculin was reduced or deleted together with vinculin. To overcome this limitation, we established metavinculin knockout mice to clarify its relevance for heart muscle pathophysiology in vivo. Our experiments revealed that the loss of metavinculin alone does not impair heart muscle development and function under physiological and the here tested pathological conditions. Even though the power of mouse models to study cardiac diseases is limited[52], these results show that metavinculin plays, if at all, a modest role in heart muscle diseases, when compared to established cardiomyopathy genes like titin[53] or desmoplakin[54]. This observation agrees with our in vitro experiments, in which we observed very clear and consistent differences in force transduction that, however, do not coincide with any obvious morphological changes of cells under steady state conditions. We, therefore, propose that future studies focus on a more subtle function of metavinculin as a modulator of cell adhesion mechanics. The set of metavinculin TSs and the mouse model generated in this study will greatly facilitate such experiments.

## Methods

**Antibodies and reagents**. The following primary antibodies were used at the indicated dilutions for immunofluorescence staining (IF) and Western blotting (WB): mouse anti-actin (sarcomeric) (Sigma, A2172; WB 1:1000), rabbit anti-α-catenin (Sigma, C2081; IF: 1:200–400; WB: 1:4000), rabbit anti-β-catenin (Sigma, C2206; IF: 1:400; WB: 1:4000), rabbit anti-connexin 43 (Cell Signaling Technologies, 3512; IF: 1:400, WB: 1:4000), rabbit anti-dystrophin (Abcam, ab15277; IF: 1:200, WB: 1:500), rat anti-integrin β1 (MB1.2, Millipore, MAB1997; IF: 1:400),

mouse anti-integrin β1d (2B1, Abcam, ab8991; WB: 1:1000), rabbit anti-FAK (Millipore, 06-543; WB: 1:1000), rabbit anti-ILK (Cell Signaling Technologies, 3862; WB: 1:1000), mouse anti-GFP (Sigma, G1546; WB:1:1000), mouse anti-N-cadherin (3B9, Thermo Fisher Scientific, 33-3900; IF: 1:500, WB: 1:2000), mouse anti-paxillin (BD Transduction Laboratories, 610051; IF: 1:200–400; WB: 1:1000), mouse anti-talin-1 (97H6, Bio-Rad, MCA4770; WB: 1:1000), mouse anti-talin-2 (68E7, Abcam, ab105458; WB: 1:2000), mouse anti-tubulin (DM1A, Sigma, T6199; WB: 1:3000), and mouse anti-vinculin (hVIN-1, Sigma, V9131; IF: 1:400, WB: 1:4000). The following secondary antibodies were used at the indicated dilutions: anti-mouse IgG Alexa Fluor-405 (Invitrogen, A31553; IF: 1:500), anti-rabbit IgG Alexa Fluor-405 (Invitrogen, A31556; IF: 1:500), anti-rabbit IgG Alexa Fluor-488 (Invitrogen, A21441; IF: 1:500), anti-rat IgG Alexa Fluor-488 (Invitrogen, A11006; IF: 1:500), anti-mouse IgG Alexa Fluor-568 (Invitrogen, A11004; IF: 1:500), anti-rabbit IgG Alexa Fluor-568 (Invitrogen, A11036; IF: 1:500), anti-mouse IgG Alexa Fluor-647 (Invitrogen, A21235; IF: 1:500), anti-mouse IgG HRP (BioRad, 170-6516; WB: 1:10,000), and anti-rabbit IgG HRP (BioRad, 170-6515; WB: 1:10000). Alexa Fluor-647 phalloidin (Invitrogen, A22287; 1:200) was used to visualize f-actin.

**Construct generation**. TS constructs were based on the human vinculin cDNA sequence NM_003373 (https://www.ncbi.nlm.nih.gov/nuccore/NM_003373). The metavinculin insert was based on sequence NM_014000 (https://www.ncbi.nlm.nih.gov/nuccore/NM_014000). Our published TS modules[5,18,19]—YPet-F40-mCherry (Addgene, 101252), YPet-FL-mCherry (Addgene, 101170), YPet-HP35-mCherry (Addgene, 101250), YPet-HP35st-mCherry (Addgene, 101251)—were inserted into the proline-rich region after aa 883 of (meta)vinculin and flanked by LE and AAA amino acid linkers. No-force control constructs were terminated directly after the TS modules, therefore lacking aa 884–1066(1134). Single-fluorophore fusion proteins were generated by incorporating YPet, mCherry, venus (A206K) or TagBFP cDNA, internally or C-terminally, as shown schematically in Supplementary Figs. 1, 3, and 7. The force-insensitive (meta)vinculin TS controls were based on a TS module with GPGGAGP (F7) linker. Mutations in (meta) vinculin cDNA were introduced using NEBuilder® HiFi DNA Assembly Master Mix (New England Biolabs, E2621L). All constructs were assembled into a modified retroviral expression plasmid pLPCX (Clontech, 631511). The correct sequence was confirmed by DNA sequencing (Eurofins Genomics). The talin-2 TS construct was described before and contains the YPet-HP35-mCherry TS module inserted after aa 450 (between the head and tail domain) and flanked by E and AA amino acid linkers. The no-force and donor-only lifetime controls contain the TS module or YPet at the C-terminus, respectively, separated by a GAAAG amino acid linker[18].

**Cell culture and construct expression**. Vinculin-deficient mouse embryonic fibroblasts (vinc$^{(-/-)}$) and its parental cell line (vinc$^{(f/f)}$)[18,24], as well as talin-deficient mouse kidney fibroblasts (tln1$^{-/-}$tln2$^{-/-}$)[18,30] and HEK293 cells (AmphoPack 293 cell line, Clontech—Takara Bio Europe, 631505), were cultured in high glucose DMEM-GlutaMAX™ medium (Thermo Fisher Scientific, 31966047) supplemented with 10% fetal bovine serum (FBS, Thermo Fisher Scientific, 10270-106) and 1% penicillin/streptomycin (P/S; Thermo Fisher Scientific, 15140122). HL-1 cells (Merck, SCC065) were cultured in Claycomb medium (Merck, 51800C) supplemented with 2 mM glutamine (Merck, G7513), 10% FBS (Merck, TMS-016-B), 0.1 mM norepinephrine (Merck, A0937-1G) and 100 μg/ml P/S (Thermo Fisher Scientific, 15140122). Constructs were expressed by transient transfection using 3 μg DNA and Lipofectamine 2000 (Thermo Fisher Scientific, 11668019) or 2–3 μg DNA and Lipofectamine 3000 (Thermo Fisher Scientific, L3000015) in a P/S-free medium. Stable cell lines were generated using the Phoenix cell transfection system[28] and selected using 2 μg/ml puromycin.

**Actin co-sedimentation assay**. This assay was performed with slight modifications according to a previously established protocol[26]. In brief, a confluent 10 cm dish of HEK293 cells was transfected with either V-V, M-V, V-TS, or M-TS using calcium phosphate precipitation[28]. The next day, cells were mechanically detached in phosphate-buffered saline (PBS), pelleted and resuspended in 1 ml of ice-cold hypotonic lysis buffer (20 mM Tris, pH 7.5, 2 mM MgCl$_2$, 0.2 mM EGTA, 0.5 mM ATP, 0.5 mM DTT) containing a protease inhibitor cocktail (cOmplete ULTRA, mini, EDTA-free EASYpack, Roche, 5892791001). After 20 min incubation on ice, cells were lysed with a Dounce homogenizer, and lysates were cleared by centrifugation at 16,000$g$ for 10 min. The total supernatant (T) was supplemented with 100 mM KCl and 5 μM actin (Sigma, A2522) and incubated in the presence or absence of 1 μM recombinantly expressed GST-IpaA[55] for 20 min on ice; GST-IpaA was expressed in *Escherichia coli* and purified from inclusion bodies by dialysis according to a previously published protocol[26]. Samples were then ultra-centrifuged at 135,000$g$ for 30 min (TLA-110 rotor, Beckman-Coulter). The soluble fraction (S) was collected, while the pellet fraction (P) was once washed in hypotonic lysis buffer and then resuspended in SDS-loading buffer. Two percent of the total (T) and soluble (S) fractions, and 10% of the pellet (P) fraction were subjected to sodium dodecyl sulphate-polyacrylamide gel electrophoresis (SDS-PAGE) and Western blotting. Note that the processing of mCherry-containing TS constructs

(V-TS and M-TS) for SDS-PAGE analysis leads to a partial fragmentation of the protein, as described before for DsRed-derived fluorophores[56].

**Immunostaining and immunohistochemistry.** For immunostainings, cells were seeded on fibronectin (FN)-coated glass slides (Menzel, #1.5) and allowed to spread for the indicated time. After fixation in 4% paraformaldehyde (PFA) for 10 min, cells were incubated in blocking buffer (2% bovine serum albumin (BSA) and 0.1% Triton X-100 in PBS). Primary and secondary antibodies were diluted in blocking buffer at the indicated concentrations (see above). Stained cells were mounted with Prolong Gold (Thermo Fisher Scientific, P36934) and images were acquired with confocal laser scanning microscopes (Leica TCS SP5 X with Leica Application Suite Advanced Fluorescence, version 2.7.3.9723, or Zeiss LSM 880 with ZEN Software, black edition) using 63× objectives (oil, NA 1.4). Cellular eccentricity was analyzed after treating live cells with a cell-permeant dye (CellMask Deep Red Plasma Membrane Stains, Invitrogen Cat. No. C10046). HL-1 cells were allowed to spread overnight, and images were acquired on a Zeiss LSM780 confocal scanning microscope (ZEN 2.1 software, version 11.0) using a 40× objective (oil, NA 1.4). For immunohistochemistry, tissue samples were isolated and fixed for at least 2 h in 4% PFA in PBS, incubated overnight in PBS with 30% sucrose (cryoprotection), and embedded in Shandon Cryomatrix (Thermo Fisher Scientific, 6769006). To unmask antigens, 5 μm sections were treated with citrate buffer (1.8 mM citric acid, 8.2 mM sodium citrate, 0.05% Tween20) for 6–12 min in a microwave. After permeabilization in 0.2% Triton X-100 for 20 min, samples were blocked with 5% BSA in PBS for 1 h and incubated with primary (overnight at 4 °C) and then secondary (2 h at room temperature) antibodies diluted in 1.5% BSA in PBS. Finally, sections were stained with DAPI and mounted with fluorescence mounting medium (Elvanol). Images were acquired using a Zeiss LSM780 confocal scanning microscope equipped with 20× objective (air, NA 0.8) for ICD and GJ markers, and 40× objective (oil, NA 1.4) for costamere markers. Images were processed using Fiji software[57].

**Fluorescence recovery after photobleaching.** To characterize FA dynamics, FRAP experiments were performed as described before[18,19]. In brief, cells were seeded on Y-shaped FN-coated micropatterns (CYTOO, 10-011-10-18) and imaged 4–6 h after seeding using a Leica SP8 confocal laser scanning microscope (LAS X Software; version 3.5.5.19976) equipped with a 63× objective (HCX PL APO, water, NA 1.2) and a 37 °C heating chamber. Two pre-bleached images were recorded at 514 nm for venus-tagged proteins and 405 nm for TagBFP-tagged proteins, followed by bleaching of a selected FA with 100% laser power for 1 s. Depending on the analyzed protein, fluorescence recovery was recorded for 300–500 s at 20 s intervals. For data analysis, the bleached region was compared with a control FA to correct for gradual bleaching during image acquisition. The FRAP profiler java plug-in for Fiji software was used to extract raw fluorescence recovery curves (http://worms.zoology.wisc.edu/research/4d/4d.html); normalized FRAP curves were plotted and fit with a single exponential in MATLAB assuming a reaction-dominated model[58].

**Immunoprecipitation, tissue lysis, and Western blot analysis.** Immunoprecipitation was performed using the μMACS GFP and HA Isolation Kit (MACS Miltenyi Biotec, 130-091-125, and 130-091-122). Cells were allowed to adhere for 2 h, washed with PBS, and lysed in 10 mM Tris/HCl (pH 7.6), 150 mM NaCl, 0.5 mM EDTA, and 0.5% NP-40 containing a protease inhibitor cocktail (cOmplete ULTRA, mini, EDTA-free EASYpack, Roche, 5892791001). Lysates were centrifuged for 10 min at 4 °C and the supernatant was incubated for 45 min with μMACS microbeads. The sample was then processed according to the instructions of the manufacturer. To analyze protein expression in tissues, samples were homogenized with an ULTRA-TURRAX T8 disperser (IKA) and lysed in 50 mM Tris/HCl (pH 7.5), 150 mM NaCl, 1% Triton X-100, 0.2% SDS, and protease inhibitor cocktail; the protein concentration was determined with BCA Protein Assay Kit (Merck, 71288). SDS-PAGE and Western blotting were performed according to standard procedures.

**Fluorescence lifetime imaging microscopy.** For live-cell FLIM analysis, cells were seeded on FN-coated (10 μg/ml, Merck, 341631-5MG) glass bottom imaging dishes (Ibidi, 81158) and allowed to spread for 4–7 h. Before imaging, the medium was changed to DMEM without phenol red (Thermo Fisher, 21063045) supplemented with 10% FBS. For FLIM of cells stably expressing T2-Con or T2-TS and transiently expressing TagBFP-tagged vinculin or metavinculin, cells were allowed to spread for 20–30 h. Transfected tln1−/−tln2−/− cells were seeded on glass bottom dishes coated with poly-L-Lysine (pLL, 0.1% (w/v), Sigma, P4707) for 15–240 min and treated with 10 μM Y-27632 inhibitor for 30 min before imaging. HL-1 cells transiently expressing sensor constructs were allowed to spread overnight, fixed in 4% PFA for 10 min to avoid artifacts caused by twitching and imaged in PBS. FLIM experiments were performed as described before[18,19,59] using two confocal laser scanning microscopes. The first system (Leica TCS SP5 X with Leica Application Suite Advanced Fluorescence, Version 2.7.3.9723, and Inspector Pro, LaVision) was equipped with a pulsed white light laser (80 MHz repetition rate, NKT Photonics), a band-pass filter for YPet (545/30 nm, Chroma), a FLIM X16 time-correlated single-photon counting (TCSPC) detector (LaVision Biotech), a 63×

objective (HCX PL APO CS, water, NA 1.2), and a heating chamber (37 °C, 5% $CO_2$; Ibidi). FA images were acquired with a scanning velocity of 400 Hz over 123.02 μm × 123.02 μm area (512 × 512 pixels); AJs were imaged over 61.51 μm × 61.51 μm area. The second confocal scanning microscope (Zeiss LSM 880 with ZEN Software, black edition, and SymPhoTime 64 software, PicoQuant) was equipped with a pulsed laser for excitation at 510 nm (LDH-D-C-510, 40 MHz repetition rate), a FLIM module from PicoQuant (MultiHarp 150 4N), a 63× objective (glycerin, NA 1.2), and a 37 °C heating chamber. FA images were acquired over 122.68 μm × 122.68 μm area (512 × 512 pixels). For each experimental condition, 30–80 images were recorded on 2–6 days.

**FLIM–FRET analysis.** Analysis of TCSPC-FLIM data were performed using custom-written MATLAB routines[18,19,59]. In brief, a multi-Otsu thresholding algorithm was applied to isolate FA specific signals using the donor intensity image; regions smaller than ~0.5 μm² were excluded from the analysis. The AJ signal was extracted by blurring the intensity image (Gaussian, $\sigma = 3$ pixels) and isolating connected bright regions. Images with inefficient signal masking were excluded manually from the analysis. The fluorescence lifetime was determined by fitting an exponential decay to the photon count time trace of each masked cell using MATLAB's 'fmincon' with a maximum-likelihood cost function based on Poisson statistics. To minimize the contribution of the instrument response function and auto-fluorescence and to ensure comparability of the resulting lifetimes, fitting was started 0.56 ns after the maximum photon count, the fit length was fixed to 9.6 ns, and lifetimes were required to have a relative fit error of <10%. The FRET efficiency (E) was calculated from the lifetime of the donor in presence of an acceptor ($\tau_{DA}$) and the average donor-only lifetime ($\tau_D$) according to Eq. (1):

$$E = 1 - \frac{\tau_{DA}}{\tau_D}. \tag{1}$$

The average donor-only lifetime was determined independently for each data set as the median of donor-only lifetimes from 4 to 6 experimental days.

To determine molecular engagement ratios, we adapted an amplitude-weighted bi-exponential fitting algorithm included in the SymPhoTime 64 analysis software (PicoQuant) and processed the data with our previously published engagement ratio analysis[19]. In short, we assume that the signal is comprised predominantly of two lifetimes: the lifetime of the TS with FRET ($\tau_{FRET}$) and without FRET ($\tau_{noFRET}$). These lifetimes can be approximated independently by control FLIM experiments: $\tau_{noFRET}$ corresponds to the donor lifetime and $\tau_{FRET}$ is determined as the short lifetime in a bi-exponential fit of the no-force control. The ratio of engaged vs non-engaged molecules was then estimated from the relative number of photons emitted by the open and closed sensor molecules. To this end, FLIM data were bi-exponentially fitted with the fixed lifetimes $\tau_{noFRET}$ and $\tau_{FRET}$ and the ratio was then rescaled to the no-force control to correct for non-fluorescent acceptor fluorophores as described previously[60].

**Generation of the metavinculin knockout mouse strain.** A homologous recombination approach was used to facilitate the deletion of the metavinculin exon from the murine Vcl gene. The targeting construct was based on clone WTSIB741I19227Q of the BAC library generated from AB2.2 ES cell DNA (129S7/SvEvBrd-Hprt[b-m2])[61]. The metavinculin-specific exon was flanked by loxP sites and a neomycin resistance cassette was included in the construct to enable selection of the successfully targeted R1 ES cell clones, which were then identified by Southern blot screening. Modified ES cells were injected into C57BL/6N host blastocysts to obtain chimera males, which were bred with Cre transgenic females (Tg(Nes-cre)1Wme)[62] to remove the metavinculin exon and the selection cassette in the F1 generation (Supplementary Fig. 5a). Cre transgene was subsequently bred out by crossing heterozygous males (M$^{(+/-,\, cre)}$) with C57BL/6N wild-type females. Genotyping of offspring was performed by three-primer PCR using the following oligonucleotides: CCGAGGTGTAGGGGTTTTCACTGC (green), AATGG-CATGCTCTCCAGGAGC (yellow), and GGAGCCAAGCAAAGCTCAGTGG (purple) (Supplementary Fig. 10a, b).

Mice were generated and housed under SPF barrier conditions at the animal facility of the Max Planck Institute of Biochemistry in Martinsried, Germany (room temperature: 22 ± 1.5 °C, relative humidity: 55 ± 5%, lighting: artificial with a light:dark cycle of 14:10 hours). All experiments involving animals were performed in accordance with animal welfare laws and were approved by the Government of Upper Bavaria (55.2-1-54-2532-77-2015).

**Transverse aortic constriction.** TAC was performed as described previously[45], with small adaptations, using 8-week-old male mice. In brief, 1 h before intubation mice received buprenorphine and metamizole intraperitoneally and were anesthetized with isoflurane. Thoracotomy was performed between the second and third rib and the diameter of the aortic arch was reduced by 65–70% by a ligature over a 27 G cannula. Mice remained in a warmed cage for 2–4 h under supervision until complete recovery from anesthesia. Sham-operated mice were treated likewise, excluding ligation of the aorta during the surgical procedure. To assess cardiac dimensions and function, pulse-wave Doppler echocardiography using a Vevo Imaging System (Fujifilm VisualSonics, VevoLab Software) was carried out before TAC/sham surgery and 4 weeks after the operation, directly before organ harvest.

**Tissue isolation and histopathology**. Tissues for histology were fixed in 4% PFA overnight and embedded in paraffin. To visualize tissue morphology, 6 μm sections were treated with hematoxylin and eosin (following standard protocol) and imaged on an Axioskop (Zeiss; SPOT v5.1). Collagen deposition was stained with Sirius Red and Fast Green. Sections of the left ventricle were imaged on a Zeiss Observer Z1 (Zeiss) with a 10× objective and interstitial fibrosis was determined with MetaMorph 7.7.1.0 as the percentage of Sirius Red positive area excluding vessels, endo- and epicardium.

**RT-qPCR**. RNA was extracted from snap-frozen left ventricular tissue samples with PureLink RNA Mini Kit (Ambion Life Technologies, 12183025) and 500 ng were reverse-transcribed with the iScript cDNA Synthesis Kit (Bio-Rad Laboratories, 1708890). The qPCR analysis was performed on 0.5 μl cDNA (in triplicates) using iQ SYBR Green Supermix (BioRad, 170-8880) and a Light Cycler 480 (Roche; LightCycler 480 Software, version 1.5). Primer sequences are listed in Supplementary Table 2. Gene expression was quantified using the 2-δδ-CT method[63] and normalized to the constitutively expressed housekeeping gene *RPL32*.

**Statistical analysis and reproducibility**. FLIM data are plotted in boxplots generated using MATLAB's 'boxplot'; the data show the median, the 25th and 75th percentile, and whiskers reaching the last data point within 1.5× interquartile range corresponding to 2.7 standard deviations for normally distributed data. The statistical significances of FLIM and cellular eccentricity data were compared by a two-sided Kolmogorov–Smirnov test with a default significance level of $\alpha = 0.05$. Data in bar graphs are presented as mean ± standard deviation. Multiple group comparisons for mouse data were performed using GraphPad Prism software package (version 6) by two-way analysis of variance followed by Sidak's multiple comparison test. Statistical significances are indicated by the $p$ value: ***$p < 0.001$; **$p < 0.01$; *$p < 0.05$; n.s. (not significant): $p \geq 0.05$. The chi-square test was used to determine if heterozygote mice breed at Mendelian ratio. Shown immunostainings and live cell images are representative of at least 2–3 independent experiments. In the case of FLIM experiments, at least 30–80 individual cells were recorded and, to ensure reproducibility, experiments were repeated on 2–6 independent days. TAC was performed on ten mice per operation- and genotype. All IF and histological analysis on heart tissue sections was performed on at least three different regions of at least three different mice per condition. Formal sample size calculation was not performed but the sample size was kept similar between experimental conditions.

**Computational codes**. TCSPC-FLIM analysis was performed with a modified version of the previously published custom-written MATLAB routine[18,19,59]; available on request.

**Reporting summary**. Further information on research design is available in the Nature Research Reporting Summary linked to this article.

## Data availability
Data supporting the findings of this paper are available from the corresponding author upon reasonable request. A reporting summary for this Article is available as a Supplementary Information file. Source data are provided with this paper.

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

## Acknowledgements

We thank Dr. Markus Moser and the animal facility of the MPI of Biochemistry for help with knockout mice generation; Julia Kerler and Anton Bomhard for performing mouse surgery and echocardiography; and Sabine Brummer for cardiac histology. This work was supported by grants of the German Research Foundation to C.G. through the Collaborative Research Consortium SFB 863 (INST 95/1209-3, GR3399/4-1) and INST 211/861-1.

## Author contributions

V.K. planned and performed cell culture experiments, mouse experiments, and analyzed the data. C.K. initiated the project with C.G., planned and performed cell culture experiments, developed data analysis routines, and analyzed the data. A.-L.S. developed the data analysis routines, supported molecular engagement ratio analysis and the data interpretation. C.B. performed HL-1 cell culture experiments and data analysis. L.W. developed data analysis routines. A.Ch-G. designed the knockout strategy, generated the knockout mice, and performed initial mouse experiments. D.R. and S.E. designed the TAC experiments, supervised surgeries, and collected/analyzed echocardiography data. C.G. initiated the study, performed, and supervised experiments. C.G. and V.K. wrote the paper with input from all the authors.

## Funding

## Competing interests

The authors declare no competing interest.
