## [Peer Review File · Nature Communications]

REVIEWER COMMENTS

Reviewer #1 (Remarks to the Author):

The authors analyze the contribution of metavinculin to force transduction in cell adhesion sites (FA and AJ) both at the molecular level and in its relevance to function of the heart muscle in the TAC model of hypertrophy using a newly developed metavinculin knockout mouse.

The manuscript is at the highest technical standard, with adequate statistics and complete documentation of methods and controls.

This is a great read with a somewhat disappointing answer to a longstanding discussion regarding the physiological relevance of metavinculin function - special and specific but not indispensable or unique.

Thanks for the privilege and pleasure of reviewing a perfect manuscript.

Wolfgang Ziegler, Hannover Medical School

small comment:

although commonly in use, a dye is not cell-permeable (only the barrier is permeable), the dye is cell-permeant (line 268)

Reviewer #3 (Remarks to the Author):

Kanoldt and colleagues outline differences in force transmission of metavinculin compared to vinculin. Using tension sensors they demonstrate that metavinculin experiences higher molecular forces in cells but is less frequently engaged to forces in comparison to vinculin. By generating specific knockout mice the authors showed that the loss of metavinculin has no consequences in vivo. Some aspects of the study are interesting but overall the study provides little novelty. Moreover, justification of stated outcomes need more rigid controls.

Main points:

- 1) It remains unclear how the lower mobility (FRAP) and the stronger engagement with talin (Co-IP) of metavinculin in comparison to vinculin can be explained with the interpretation the authors provide with the tension sensors. Increased binding of metavinculin to talin would lead to the assumption that it engages better with actin, at least under the assumption that talin binding influences the exposure of the actin-binding site. The decrease in mobility would be in line with this interpretation but would oppose the interpretation of data with tension sensors. Different interpretations would need additional sets of experiments involving mutants that influence engagement with talin and actin (see below).
- 2) Metavinculin has increased talin binding, but the talin has reduced tension. So, is talin2 engaging less with actin? It will be difficult to conclude this without biochemistry to assess talin2-actin binding in the presence of vinculin or metavinculin. It is difficult to see how the inclusion of the metavinculin splice insert in the vinculin tail would influence talin-actin binding.
- 3) From the tension sensor data provided, it seems that metavinculin (Fig 2g) engages less than vinculin in every force range tested. Additional mutations that alter the engagement with talin (e.g. A50I mutation) and actin (constitutively active forms) are needed to provide more substantial support for the interpretation of differences seen in engagement with actomyosin.

4) How can the authors be certain that the different FRET efficiencies of the two vinculin isoform tension sensors are not due to differences of the intrinsic rotational freedom of fluorophores. One can assume that the rotational freedom is restricted when proteins become activated and differences in FRET efficiencies might be rather due to differences in the proteins' structure rather than the impact of forces. The use of mutants outlined in point 2 in the presence and absence of forces may help to better explore the force contribution. Other mutants outlined in Chorev et al (Sci Rep 2018) might also help to gain more mechanistic insight.

5) The authors don't provide FLIM images and therefore it is very difficult to judge the accuracy and potential flaws of quantifications. Also, quantifications are limited to the focal adhesion area and it is not clear why authors threshold to this area. Moreover, all experiments were done using live-cell FLIM and it is not really clear why authors did not perform movies with their available constructs. Recordings of force changes in individual adhesions and influencing force regimes using drugs would potentially add significantly more mechanistic insight.

6) If vinculin and metavinculin engage at different force ranges, there may be more suitable tension sensors used by the Schwartz laboratory that cover the whole range and are not "on-off" sensors.

7) How can authors be sure that the TS insert had no effect on the activation status of the vinculin isoforms and therefore influence also force regimes? Moreover, is it possible that metavinculin and vinculin overexpression influence intracellular tension in a different manner?

8) Authors state that their observations are due to differential force engagement but experiments under force releasing conditions have not been performed. For a convincing interpretation, such experiments would need to include mutants mentioned under point 2.

9) Statements about similarities of isoform specific force transduction in FAs and AJs seem premature without a more profound assessment. Tension sensor measurements under different force conditions would help gain more insight.

10) Chorev et al. (Sci Rep 2017) studied the relation of forces and conformational changes of different vinculin forms, including vinculin, metavinculin and mutants of the two. It is worth discussing this manuscript in context with the tension sensor data. Authors should also cite and discuss the paper of B. Hoffman's group in eLife (doi:10.7554/eLife.33927) – where they did FRAP on their different vinculin tension sensor constructs that report different force ranges. Their take-home message is that once vinculin is activated (by talin or actin) it is pulled until a certain extension is reached, rather than until a set force is achieved.

11) The knock out data are on the one hand interesting but the lack of a phenotype makes the potential differences seen in the molecular behavior of metavinculin in comparison to vinculin less attractive.

Reviewer #4 (Remarks to the Author):

Metavinculin is an understudied protein and given its importance to heart disease, this manuscript makes a significant contribution. The most value that it brings to the field is the generation of the metavinculin knockout mouse. Through their knockout findings, the authors conclude that the role of metavinculin in heart disease has been overstated in the past. This is an important finding and the manuscript should focus on this and in particular compare it with the effects seen in the vinculin knockout that is embryonic lethal E10. Why is the metavinculin knockout not embryonic lethal? The remainder of the experiments are less convincing because of the choice of mutants, some missing controls, and missing details and confusing nomenclature.

Without enough information about the constructs, I find this manuscript and results difficult to follow

Some information might be available in the literature but this is difficult to follow without providing this information here that I am wondering how these FRET probes inserted into the polypeptide chains affect the structure of metavinculin; a control, that these altered polypeptide chains are still functional, for example by binding to known binding partners, is missing

The vinculin null cells seem to spread (Figures 1A, 1C, 2A) as WT cells would; how does the cytoskeleton look like, and what is the vinculin or metavinculin expression rescuing as the null cells do not look different from how WT cells usually look like

Why are HL-1 cells instead of the vinculin null cells used in Figure 3 and what vinculin or metavinculin do these mouse atrial cardiomyocytes express endogenously?

Figure S1A could be improved to be more informative about the domain structure of the two proteins and could go into the main manuscript

The overuse of acronyms distracting; the authors introduce almost an entire new language, and for example "Vad" is only defined in Figure legend 2 (line 558)

Figure S1A – please add the residue numbers in the legend, actually, I cannot see where the T1B construct is defined (same for VTS and actually all constructs), please do so in the main manuscript, in the methods, in the legends, and in the Figure

Figure S1C and S7 – please show controls, full blots, and MWM

Figures S3A-S3D - please improved to be more informative, see suggestions above

Lines 84-85 reads "The FRET control construct, which lacks the vinculin-tail domain", please be specific so that others can follow and repeat, what residue numbers etc please

Actually, I cannot see a pull down of the I997A vinculin or in particular the I1065A mutant in reference 25 that is cited; reference 31 does have such a pull down for vinculin only (not for metavinculin) in Figure 1A but there is unpolymerized actin also in the supernatant and the pellet of the actin band for WT seems less than for I997A and there is less unpolymerized actin in WT compared to I997A; either way Thompson et al. (2014) conclude that the mutant has a "significant decreased affinity for F-actin" which is according to reference #31 Figure 1C ~65% binding for WT compared to ~48% binding for I997A which means that I997A still binds about 74% compared to WT; several metavinculin (and vinculin) reconstructions with F-actin have been published, and the only mention of metavinculin I1065A I can find is in the yet to be peer reviewed 2020 study on bioRxiv by Alushin's laboratory

PDB entry 3jbc for Mvt/actin complex only has residues 917-1046 so the I1065A effects on actin remain unclear

Line 105, here too it would help to say a sentence about this (published) construct

Lines 238 onwards – really difficult to understand what was used

Line 244 – what is the "Actin-binding-deficient mutant" and how is deficient defined, please see comments above?

Figure 2G – please show the raw data for all repeats

I find Figure 1 too low resolution to see protein localization more specifically as usually one would see the focal adhesions

Can S1B and S2 please be in color?

Try and avoid the first person for a protein (please replace “vinculin’s” with “of vinculin”)

Controls and MWM seem missing for Figures 1F and 1G
Please add MWM for Figure 4A and show full blots

Please show the full blots also, not just the cropped

Response to Reviewer #1

The authors analyze the contribution of metavinculin to force transduction in cell adhesion sites (FA and AJ) both at the molecular level and in its relevance to function of the heart muscle in the TAC model of hypertrophy using a newly developed metavinculin knockout mouse. The manuscript is at the highest technical standard, with adequate statistics and complete documentation of methods and controls. This is a great read with a somewhat disappointing answer to a longstanding discussion regarding the physiological relevance of metavinculin function - special and specific but not indispensable or unique. Thanks for the privilege and pleasure of reviewing a perfect manuscript.

Response: We feel honoured to receive such encouraging comments from an internationally recognized vinculin expert. Thank you very much for the support.

small comment:

although commonly in use, a dye is not cell-permeable (only the barrier is permeable), the dye is cell-permeant (line 268)

Response: We have changed “cell-permeable” to “cell-permeant” in the new manuscript.

Response to Reviewer #3

Kanoldt and colleagues outline differences in force transmission of metavinculin compared to vinculin. Using tension sensors they demonstrate that metavinculin experiences higher molecular forces in cells but is less frequently engaged to forces in comparison to vinculin. By generating specific knockout mice the authors showed that the loss of metavinculin has no consequences in vivo. Some aspects of the study are interesting but overall the study provides little novelty. Moreover, justification of stated outcomes need more rigid controls.

Response: We thank the reviewer for taking the time to carefully evaluate our manuscript. As suggested, we generated new vinculin and metavinculin constructs and performed a range of additional experiments. We included further zero-force controls to validate that the observed effects are force-specific, we generated and analysed (meta)vinculin mutants, and we performed biochemical assays to demonstrate that the insertion of the tension sensor module into vinculin or metavinculin does not affect the activation status of these proteins. We also clarified the used nomenclature, included a further western blot control, and provide more detailed information on how the expression constructs were generated, as requested by reviewer #4. Altogether, we believe that this has further strengthened the manuscript and hope that the reviewer will be now able to fully support the publication of our study.

Main points:

1) It remains unclear how the lower mobility (FRAP) and the stronger engagement with talin (Co-IP) of metavinculin in comparison to vinculin can be explained with the interpretation the authors provide with the tension sensors. Increased binding of metavinculin to talin would lead to the assumption that it engages better with actin, at least under the assumption that talin binding influences the exposure of the actin-binding site. The decrease in mobility would be in line with this interpretation but would oppose the interpretation of data with tension sensors. Different interpretations would need additional sets of experiments involving mutants that influence engagement with talin and actin (see below).

Response: The observation that increased (meta)vinculin activation (as well as talin and actin binding) does not directly correlate with tension is one of the important findings of this study; but it is not in conflict with the published literature. There is no doubt that vinculin activation is necessary for force transduction, but it is not sufficient for mechanical loading.

Please note that this observation is consistent with our previous experiments on vinculin, in which we showed that vinculin activation and vinculin force transduction are separable events (Grashoff et al, Nature, 2010). The findings are also in line with data from other groups, for example recently published data from the Ballestrem lab showing that a tight interaction between vinculin and talin is possible in force-free conditions (Atherton et al, JCB, 2020). Thus, the observation that metavinculin is more associated with talin but not more frequently mechanically loaded, is not inconsistent with existing data. The reason for differences in molecular forces seems to be a differential engagement of the vinculin isoforms with actin filaments. This is already supported by our own experiments showing that impairing actin engagement (by I997A and I1065A mutations; Fig. 2c and Supplementary Fig. 7a) abolish vinculin isoform-specific differences. Please note that this finding is also in line with published data. Janssen et al,

(JCB, 2012), for instance, proposed that metavinculin engages but severs actin, while a follow-up study suggested that metavinculin efficiently binds to actin filaments, but makes them more susceptible to mechanical breakage (Durer et al, JMB, 2015). Moreover, structural analyses confirmed that vinculin and metavinculin tail domains engage actin networks in different ways (Kim et al, J Mol Biol, 2016).

Thus, our finding fit existing data quite well and we show, for the first time, how the increased talin binding and altered actin filament engagement of metavinculin affects force transduction in FAs: Metavinculin linkages are less frequently exposed to mechanical forces but those molecules that are exposed to tension, experience higher mechanical loads. We hope the reviewer will agree that this is a finding worth publishing.

2) Metavinculin has increased talin binding, but the talin has reduced tension. So, is talin2 engaging less with actin? It will be difficult to conclude this without biochemistry to assess talin2-actin binding in the presence of vinculin or metavinculin. It is difficult to see how the inclusion of the metavinculin splice insert in the vinculin tail would influence talin-actin binding.

Response: This point relates to the discussion above. We do not argue that the expression of metavinculin directly affects the interaction of talin-2 with actin. In fact, we also think that this is unlikely. What we show is that the vinculin-dependent modulation of talin tension, previously shown by us (Austen et al, Nat Cell Biol, 2015) and independently observed by the Schwartz group (Kumar et al, JCB, 2016), is vinculin isoform-specific. Consistent with the argument above, metavinculin appears to form actin-interactions that are less frequently exposed to tension. The binding of metavinculin to talin therefore does not contribute to an increase in talin-2 forces as much as vinculin does.

3) From the tension sensor data provided, it seems that metavinculin (Fig 2g) engages less than vinculin in every force range tested. Additional mutations that alter the engagement with talin (e.g. A50I mutation) and actin (constitutively active forms) are needed to provide more substantial support for the interpretation of differences seen in engagement with actomyosin.

Response: We believe that these mutants are not particularly helpful to control for effects on force propagation, because (meta)vinculin binding to talin and the status of (meta)vinculin activation do not directly correlate with tension (see discussion above). However, as suggested by the reviewer, we generated the A50I mutants for FL-based vinculin and metavinculin sensors and analysed these constructs together with all controls (V-TS, M-TS, Con-TS) in vinculin-deficient cells. We observed a decrease in FRET efficiency for A50I mutant vinculin and metavinculin constructs, again consistent with the notion that talin-binding and force transduction are separable. These data are now shown in Fig. 2g. Since a previous study observed a slight but significant FRET increase for a F40-based vinculin construct (Rothenberg et al, Biophys J, 2018), we generated another set of A50I mutants using F40 sensor modules. Again, we observed a decrease in FRET efficiencies for vinculin and metavinculin constructs and these data are shown in Fig. 2h.

4) How can the authors be certain that the different FRET efficiencies of the two vinculin isoform tension sensors are not due to differences of the intrinsic rotational freedom of fluorophores. One can assume that

the rotational freedom is restricted when proteins become activated and differences in FRET efficiencies might be rather due to differences in the proteins' structure rather than the impact of forces. The use of mutants outlined in point 2 in the presence and absence of forces may help to better explore the force contribution. Other mutants outlined in Chorev et al (Sci Rep 2018) might also help to gain more mechanistic insight.

Response: To provide further evidence that the observed effects are force specific and not a consequence of distinct conformational states of vinculin and metavinculin, we included two additional controls. First, we generated an additional sensor module, in which the fluorophores are separated by a very short, only seven amino acids long flagelliform sequence (called F7). This control construct does not respond to mechanical forces by a change in FRET efficiency, as the F7 peptide cannot be sufficiently extended to cause a measurable change in energy transfer rates. By contrast, changes in fluorophore orientation will be still detected as the linker is highly flexible (GPGGAGP). We inserted this F7 sensor module into the same insertion site of vinculin and metavinculin, expressed these control constructs in cells, and performed live cell FLIM experiments. The data show that FRET efficiencies for vinculin- and metavinculin-F7 controls are virtually the same demonstrating that the observed effects are not due to differences in protein structure. The data are now included in Fig. 2e.

Moreover, we inserted vinculin and metavinculin tension sensors into talin1/2 knockout cells, seeded them on poly-L-Lysine (pLL) and also treated them with Rock inhibitor (Y-27632). Under these conditions, both vinculin isoforms cannot be exposed to mechanical tension. FLIM measurements show that FRET efficiencies are highly similar for V-TS, M-TS, and Con-TS demonstrating that the observed effects are force specific. These data are now shown in Fig. 2d.

5) The authors don't provide FLIM images and therefore it is very difficult to judge the accuracy and potential flaws of quantifications. Also, quantifications are limited to the focal adhesion area and it is not clear why authors threshold to this area. Moreover, all experiments were done using live-cell FLIM and it is not really clear why authors did not perform movies with their available constructs. Recordings of force changes in individual adhesions and influencing force regimes using drugs would potentially add significantly more mechanistic insight.

Response: We have included an additional figure (now Supplementary Figure 6) to illustrate the workflow of our data analysis pipeline. This should allow the reader, together with the materials & methods section, to understand how the data were analysed. Please note that our previously published studies include detailed protocols on data analysis (e.g. Ringer et al, Nature Methods, 2017; Price et al, Nature Comm, 2018; Cost, Khalaji and Grashoff, 2019, Current Protocols in Cell Biology).

We specifically work with the FA signal because fluorescence intensities/photon counts from the cytoplasm/plasma membrane are typically too low to ensure proper FRET analyses in cells expressing the tension sensor constructs to wildtype (wt)-levels (and do not overexpress the protein).

Besides this, we respectfully disagree with the notion that showing individual FRET images is strong evidence for observed tendencies. FRET experiments are inherently noisy and the observed differences in tension sensor experiments are comparably small. We have therefore strongly advocated the use of

quantitative data analyses over the last years and would like to further emphasize that this form of analysis is preferable to showing individual images.

We did not perform movies for two reasons. First, the here used cells are rather static and not suitable for analysing processes of cell migration. Second, we do not think that the analysis of migrating fibroblasts will provide physiologically relevant insights into the function of metavinculin, which appears to be expressed in non-motile cell types.

6) If vinculin and metavinculin engage at different force ranges, there may be more suitable tension sensors used by the Schwartz laboratory that cover the whole range and are not “on-off” sensors.

Response: The Schwartz group is using the F40 sensor (originally described in Grashoff et al, Nature, 2010), which is also used in this study. The additional probes, jointly published by the group of Martin Schwartz and Taekjip Ha (Brenner et al, 2016, Nano Letters), are F25 and F50 tension sensor modules. Single-molecule force spectroscopy experiments indicate that these sensors allow force measurements at different force magnitudes. However, the single-molecule calibration in the Brenner et al study is using peptides that are associated with organic dyes (see Brenner et al. Fig. 1) resulting in higher FRET efficiencies and larger dynamic range. When inserted between GFP-like fluorophores, differences in vinculin tension are difficult to detect (see Brenner et al. Fig. 4c). Unfortunately, that specific data set lacks a statistical evaluation, but the data indicate that vinculin-F40 and vinculin-F50 sensors are indistinguishable, while the vinculin-F25 sensor shows hardly any difference to the control. We are therefore convinced that the use of those sensors would not add any additional information to our experiments. Please note that all our tension sensor modules, comprising force-sensitive linker inserted between donor and acceptor fluorophores, have been single-molecule calibrated and their FRET-force correlations are known (Austen et al, Nat Cell Biol, 2015; Ringer et al, Nature Methods, 2017). We would also like to emphasize that “on-off” sensors are ideally suited to extract engagement ratios, which are different between the vinculin isoforms and thus central to this study.

7) How can authors be sure that the TS insert had no effect on the activation status of the vinculin isoforms and therefore influence also force regimes? Moreover, is it possible that metavinculin and vinculin overexpression influence intracellular tension in a different manner?

Response: To address whether the insertion of the tension sensor module into vinculin and metavinculin affects the activation status, we performed actin co-sedimentation assays for vinculin-venus, metavinculin-venus, vinculin tension sensor and metavinculin tension sensor constructs in the presence or absence of the vinculin activating peptide IpaA. Consistent with our previous experiments (Grashoff et al, 2010, Nature, Supplementary Fig. 2c), in which we show that sensor insertion does not lead to constitutive vinculin activation, we now demonstrate that also metavinculin is largely unaffected by the insertion of the sensor module. Both vinculin and metavinculin can be shifted from the soluble fraction (S) to the pellet fraction (P) by the addition of IpaA. The new data are now shown in Supplementary Fig. 5. Please note that the processing of mCherry-containing tension sensor lysates (V-TS and M-TS) for SDS-PAGE analysis leads to a partial fragmentation of the protein, as described for DsRed-derived fluorophores before (LaGross et al, 2000, PNAS).

Regarding the concern of overexpression, we note that we do not include overexpressing cells into our analyses. As shown in Supplementary Fig. 1, we reconstituted (meta)vinculin in knockout cells to about wt-levels and we used this expression level in all experiments.

8) Authors state that their observations are due to differential force engagement but experiments under force releasing conditions have not been performed. For a convincing interpretation, such experiments would need to include mutants mentioned under point 2.

Response: As discussed above, we do not think that the suggested vinculin mutants, which induce a distinct activation status, are suitable to control vinculin force transmission. Instead, we generated force releasing conditions by expressing our constructs in talin1/2 knockout cells, seeding them on pLL-coated dishes and inhibiting ROCK activity by addition of the Y-27632 compound. This will certainly reflect a situation in which (meta)vinculin is not exposed to mechanical forces. The FLIM experiments, now shown in Fig. 2d, show that effects are indeed force-specific.

9) Statements about similarities of isoform specific force transduction in FAs and AJs seem premature without a more profound assessment. Tension sensor measurements under different force conditions would help gain more insight.

Response: We have included the HL-1 cells to demonstrate that the difference in force transduction can be also observed in a physiologically more relevant cell type, which also expresses low levels of metavinculin (now shown in Supplementary Fig. 9). The new control experiments show that vinculin isoform specific effects in force transduction are independent of conformation (Fig. 2e) and also force-specific (Fig. 2d), and vinculin isoform specific effects are reproduced in FAs of HL-1 cells (Fig. 3b). Please also note that the vinculin tension sensor has been used to study molecular mechanics in AJs before (Leerberg et al, Current Biology, 2014). We therefore hope, the reviewer will agree that further control experiments are not required.

10) Chorev et al. (Sci Rep 2017) studied the relation of forces and conformational changes of different vinculin forms, including vinculin, metavinculin and mutants of the two. It is worth discussing this manuscript in context with the tension sensor data. Authors should also cite and discuss the paper of B. Hoffman's group in eLife (doi:10.7554/eLife.33927) – where they did FRAP on their different vinculin tension sensor constructs that report different force ranges. Their take-home message is that once vinculin is activated (by talin or actin) it is pulled until a certain extension is reached, rather than until a set force is achieved.

Response: We thank the reviewer for this suggestion. The finding from Chorev et al that metavinculin exists in a semi-open conformation is consistent with our observation of metavinculin showing an enhanced interaction with talin. This is now discussed in the main text. We think our data provide the interesting perspective that vinculin and metavinculin also differ in how they carry mechanical loads.

We would like to refrain from an extended discussion of the La Croix et al manuscript. As discussed in our study and shown before (Ringer et al, 2017, Nat Methods), force transmission is regulated by cells on at least two levels: the force per molecule and the fraction of molecules being under force. The model by La

Croix et al addresses only the force-per-molecule component of the phenomenon, a useful comparison with our data is therefore hardly possible.

Besides this, we find it rather unlikely that vinculin is regulated by an extension-based mechanism; certainly, this model is not supported by our data. The degree of extension of the here used tension sensor peptides, F40 and FL, are very different and unfolding of the FL peptide will lead to a much greater increases in extension as compared to the F40 construct, because the peptide is longer (82aa vs 40 aa) and unfolds in a one-step process. Still, the observed FRET efficiency differences are highly similar (Supplementary Fig. 8). In general, we find it hard to understand how vinculin could tolerate the insertion of tension sensor module (containing two comparably large fluorophores) but be sensitive to nm-scale differences in peptide extension.

We think that discussing these issues would be rather confusing to the reader and divert from the main take home message of our study. Resolving this issue in a proper fashion would require performing a set of additional experiments using a range of tension sensor modules from the Hoffman group. We hope the reviewer will agree that this would go much beyond the scope of this manuscript.

11) The knock out data are on the one hand interesting but the lack of a phenotype makes the potential differences seen in the molecular behavior of metavinculin in comparison to vinculin less attractive.

Response: We think that reviewer #1 summarized the here identified role of metavinculin very nicely. A function that is *'special and specific but not indispensable or unique'*. We also note that the issue of rather mild phenotypes of knockout mice has been observed before and has been extensively discussed (e.g. Barbaric et al. 2007, Brief Funct Genomic Proteomic). As we point out in the manuscript, the lack of phenotype in mice is not evidence that the here observed *in vitro* effects are irrelevant. Nevertheless, the *in vivo* data provide an important insight into the patho-physiological relevance of the here described effects. Compared to established cardiomyopathy genes like titin, metavinculin clearly plays a more modest role for the development of heart muscle disorders. We are convinced that this is a very important finding that has to be communicated to the scientific community. The mechanistic insights into (meta)vinculin force transduction will be key to unravel the true molecular function of this vinculin isoform in the future.

Response to Reviewer #4

Metavinculin is an understudied protein and given its importance to heart disease, this manuscript makes a significant contribution. The most value that it brings to the field is the generation of the metavinculin knockout mouse. Through their knockout findings, the authors conclude that the role of metavinculin in heart disease has been overstated in the past. This is an important finding and the manuscript should focus on this and in particular compare it with the effects seen in the vinculin knockout that is embryonic lethal E10. Why is the metavinculin knockout not embryonic lethal? The remainder of the experiments are less convincing because of the choice of mutants, some missing controls, and missing details and confusing nomenclature.

Response: We thank this reviewer for the encouraging remarks and the positive evaluation. We agree that the description of the metavinculin knockout mouse is of particular importance. We have addressed all of the issues raised by this reviewer and included a whole range of additional experiments requested by reviewer #3. We feel that this has further improved the manuscript and we hope that the reviewer can now fully support the publication of our study.

Regarding the non-lethal phenotype of metavinculin deficient mice, we think that this observation was rather to be expected. Our observation that metavinculin is expressed at comparably low levels in most tissues is consistent with previous reports and it seems likely that certain functions of metavinculin can be compensated by vinculin, which is still expressed in metavinculin-deficient mice. In fact, such an assumption would be consistent with our *in vitro* data showing that the morphology of vinculin-deficient cells can be equally rescued with both isoforms.

Without enough information about the constructs, I find this manuscript and results difficult to follow.

Response: We added additional information in the figure and figure legend of Supplementary Figure 1 and Supplementary Figure 3, and we clarified potentially confusing nomenclature throughout the text. Since we use four different tension sensor modules that are inserted into three different molecules (vinculin, metavinculin and talin), and because reviewer #3 asked for additional controls, a certain level of complexity cannot be avoided. Nevertheless, we think that the new manuscript will be easier to follow.

Some information might be available in the literature but this is difficult to follow without providing this information here that I am wondering how these FRET probes inserted into the polypeptide chains affect the structure of metavinculin; a control, that these altered polypeptide chains are still functional, for example by binding to known binding partners, is missing.

Response: To address the issue of vinculin/metavinculin functionality, we performed an actin co-sedimentation assay in the presence and absence of the vinculin activator IpaA. As shown previously, this assay can be used to evaluate the activation status of vinculin *in vitro* (Chen et al, JCB, 2005) and was previously applied by the last author of this study to show that the insertion of the tension sensor into vinculin does not affect vinculin function (Grashoff et al, Nature, 2010). We now performed the experiment by analysing vinculin-venus, metavinculin-venus, vinculin tension sensor, and metavinculin tension sensor in parallel. These experiments show that vinculin and metavinculin can be activated by

IpaA to engage with actin, demonstrating that the insertion of the tension sensor module does not lead to a constitutive activation of the target molecules. These data are now shown in the new Supplementary Figure 5. Please note that the processing of mCherry-containing tension sensor constructs (V-TS and M-TS) for SDS-PAGE analysis leads to a partial fragmentation of the protein, as described for DsRed-derived fluorophores before (LaGross et al, 2000, PNAS).

The vinculin null cells seem to spread (Figures 1A, 1C, 2A) as WT cells would; how does the cytoskeleton look like, and what is the vinculin or metavinculin expression rescuing as the null cells do not look different from how WT cells usually look like.

Response: Overall, loss of vinculin has a comparably small effect on the cell morphology, for example when compared with the phenotype of talin-depleted cells (Austen et al., Nat Cell Biol, 2015). As stated in the main text, the here used vinculin-deficient cells display a reduced cellular eccentricity 2 h after cell adhesion, meaning cells adopt a more roundish morphology than their wildtype counterparts during cell spreading. This has been quantified in Fig. 1b and Fig. 2b. To further illustrate this point, we acquired additional immunostainings of such cells at 2h after spreading, now shown in the new Supplementary Fig. 4.

Why are HL-1 cells instead of the vinculin null cells used in Figure 3 and what vinculin or metavinculin do these mouse atrial cardiomyocytes express endogenously?

Response: The HL-1 cells are used to confirm the differential role of the vinculin isoforms in a distinct cell type that is physiologically more relevant in terms of cardiac function than embryonic fibroblasts. Mouse embryonic fibroblasts do not produce metavinculin (Suppl. Fig.1c), while HL-1 cells are derived from mouse atrial cardiomyocytes, which do express this vinculin isoform. A western blot of HL-1 cell lysates is now shown in the new Supplementary Figure 9.

Figure S1A could be improved to be more informative about the domain structure of the two proteins and could go into the main manuscript.

Response: We now include more detailed information on the domain structure of the protein and also explain the construct design in more detail in the figure legends as well as in the main text. Since the construction of the vinculin tension sensor has been shown schematically many times before (e.g. Grashoff et al, Nature, 2010), we would prefer to show this drawing in the supplement and use the limited space of the main figures for primary data.

The overuse of acronyms distracting; the authors introduce almost an entire new language, and for example “Vad” is only defined in Figure legend 2 (line 558).

Response: We are now clearer when introducing any acronyms and also provide more detailed information about the acronyms in Supplementary Figures and the respective figure legends. We agree with the reviewer that naming the I997A and I1065A constructs “Vad” mutations was somewhat confusing. We clarified this also in the main text and figure legends. These constructs are now named: V-TS-I997A and M-TS-I1065A.

Figure S1A – please add the residue numbers in the legend, actually, I cannot see where the T1B construct is defined (same for VTS and actually all constructs), please do so in the main manuscript, in the methods, in the legends, and in the Figure.

Response: We now specifically indicate the residue numbers of the insertion sites in main text and the figure legends. Tension sensor modules were inserted after aa 883 in (meta)vinculin and after aa 450 in talin-2.

Figure S1C and S7 – please show controls, full blots, and MWM

Response: The full blots are shown in the source data. We now include a statement in the respective figure legends that refers to these original data sets. Molecular weights are indicated.

Figures S3A-S3D - please improved to be more informative, see suggestions above

Response: We now include more detailed information into the new Supplementary Fig. 3 and the figure legend. Please note that detailed information is also available in the methods section. Altogether, this should help the reader to understand how the individual constructs were designed.

Lines 84-85 reads “The FRET control construct, which lacks the vinculin-tail domain”, please be specific so that others can follow and repeat, what residue numbers etc please.

Response: We specified this sentence and also include residue numbers into the relevant descriptions. Specifically, we state: “*In parallel, we generated control constructs to determine the fluorescence lifetime of the donor fluorophore as well as the FRET efficiency of the no force control (Con TS), which comprises the vinculin head domain (aa 1–883) and a tension sensor (TS) module but lacks the vinculin tail domain (Supplementary Fig. 3a c).*”

Actually, I cannot see a pull down of the I997A vinculin or in particular the I1065A mutant in reference 25 that is cited; reference 31 does have such a pull down for vinculin only (not for metavinculin) in Figure 1A but there is unpolymerized actin also in the supernatant and the pellet of the actin band for WT seems less than for I997A and there is less unpolymerized actin in WT compared to I997A; either way Thompson et al. (2014) conclude that the mutant has a “significant decreased affinity for F-actin” which is according to reference #31 Figure 1C ~65% binding for WT compared to ~48% binding for I997A which means that I997A still binds about 74% compared to WT; several metavinculin (and vinculin) reconstructions with F-actin have been published, and the only mention of metavinculin I1065A I can find is in the yet to be peer reviewed 2020 study on bioRxiv by Alushin’s laboratory

PDB entry 3jbk for Mvt/actin complex only has residues 917-1046 so the I1065A effects on actin remain unclear.

Response: To our knowledge metavinculin-I1065A has not been investigated previously (besides the article on bioRxiv, mentioned by the reviewer). As the I997A mutation clearly impairs actin binding (e.g. Thievensen et al, 2013, JCB) and was previously used to control for vinculin tension sensor experiments (Rothenberg et al, 2018, Biophys J), we inserted the same isoleucine to alanin mutation at the

corresponding residue in metavinculin. The underlying assumption is that – due to the amino acid conservation of this region in vinculin and metavinculin – the actin-binding function is maintained. This assumption is in line with the tension sensor experiments in Fig. 2c showing the expected FRET efficiency increase to the zero-force level.

Line 105, here too it would help to say a sentence about this (published) construct

Response: As requested, we expanded the description of the talin-2 tension sensor.

Lines 238 onwards – really difficult to understand what was used

Response: We tried to clarify this. Please note that the mentioned constructs are available on Addgene, where detailed electronic maps of all constructs can be found. Thus, the interested reader will be able to find any information of these published constructs.

Line 244 – what is the “Actin-binding-deficient mutant” and how is deficient defined, please see comments above?

Response: We now more precisely state that the mutations are reducing actin-binding affinity and refer to the respective studies (Thievensen et al, JCB, 2013; Thompson et al, Structure, 2014). As indicated above, the constructs were renamed (V-TS-I997A and M-TS-I1065A) to be as specific as possible.

Figure 2G – please show the raw data for all repeats

Response: The raw data of this representation are shown in Fig. 2i and 2j. We added this information into the figure legend of Fig. 2k. Please note that Fig. 2 includes more data sets now and the numbering has been changed.

I find Figure 1 too low resolution to see protein localization more specifically as usually one would see the focal adhesions

Response: We could not quite understand this suggestion, because the image shown in Fig. 1 has been acquired with a state-of-the-art confocal microscope using 63x and additional zoom to display the overlay of (meta)vinculin-venus and paxillin signals. Please note that the scale bar in the zoom-ins indicate 5 μm , and structure much smaller than that are still resolved. This is as good as it gets with a diffraction-limited microscope.

Can S1B and S2 please be in color?

Response: We find that the relevant structures can be appreciated much better in the grey scale mode, especially when the figures are printed. For comparison, we have included the greyscale and colour figures side-by-side at the end of this response. We would prefer to keep the figures in grey scale and hope the reviewer will agree with our choice.

Try and avoid the first person for a protein (please replace “vinculin’s” with “of vinculin”)

Response: This has been changed.

Controls and MWM seem missing for Figures 1F and 1G

Response: The controls are indicated by IB (immunoblot), while the samples from the immunoprecipitation are indicated as IP. The molecular weights of all proteins are now indicated.

Please add MWM for Figure 4A and show full blots

Please show the full blots also, not just the cropped

Response: Molecular weights in Fig. 4a are now indicated and the full blots are shown in the source data. We now include a statement in the figure legends that refers to these original data sets.

Figure 1 for reviewer #4: Comparison of Supplementary Fig. 1b colour vs. grey scale.

Legend Figure 1: Parts of Supplementary Fig. 1b shown in colour and greyscale. We find it easier to distinguish subcellular structures (especially for the actin staining) in greyscale.

Figure 2 for reviewer #4: Comparison of Supplementary Fig. 2 colour vs. grey scale.

Legend Figure 2: Supplementary Fig. 2 shown in colour and greyscale. Due to the low fluorescence intensity of the talin-1-TagBFP signal, structures seem easier to observe in the greyscale mode.

Figure 3 for reviewer #4: Comparison of Supplementary Fig. 3 colour vs. greyscale.

Legend Figure 3: Supplementary Fig. 3d shown in colour and greyscale. In our view, subcellular structures are easier to distinguish in the greyscale mode, especially when the figures is printed and viewed on hardcopy.

REVIEWER COMMENTS

Reviewer #3 (Remarks to the Author):

I am very happy about the thorough revision and additional controls. They have improved the quality of the manuscript significantly and I therefore recommend publication.

Reviewer #4 (Remarks to the Author):

I find the apparent addition of the construct information to the Figure legends insufficient; their explanation as to why not describe these in the main manuscript, i.e. the methods section in detail also, is unclear as it would be in everyone's interest to follow the experiments easily:

Figure legend 1a-1c, 1e

The tag is suggested to be on the N-terminus from the methods but on the C-terminus in Supplementary Figure 1a; what about the linker region and how is it ensured that the tag does not hinder or force (meta)vinculin opening to its active conformer? This is perhaps answered for the TS constructs in Supplementary Figure 5

Figure legend 1d

Same comment for talin constructs as for vinculin in 1a-1c, 1e

Figure legend 1f-1g

Please show all markers (the lane is missing with all, not just the nearest to the protein of interest, and what standard is at 151 kDa for example, sorry for not explaining better before) as well as the controls

Figure legend 2

Please describe the constructs in full

The insertion at residue 883, thank you for now mentioning this, might probably affect its interaction with the helix bundle

I cannot find the description for V-TS or M-TS (panels a and c in Supplementary Figure 3 do not have enough details and the reference (or information available on Addgene) does not replace a one sentence description of the constructs; "lacking the C-terminal actin binding domain" is which residues please? "In parallel, we generated control constructs to determine the fluorescence lifetime of the donor fluorophore as well as the FRET efficiency of the no force control (Con TS), which comprises the vinculin head domain (aa 1-883) and a tension sensor (TS) module but lacks the vinculin tail domain (Supplementary Fig. 3a c)." meaning 884-1061 were deleted?) or FL-based tension sensors or the two F7-TS constructs, or Ts-TS or the two B constructs (panel a in Supplementary Figure 1 does not have enough information) etc

Figure 3 legend

Same as for Figure 2 and also not clear how the Ypet fusion is constructed

Figure 4

Please show the entire blot (some, for example tubulin, still seem cropped in the source data) including the lanes with the markers as well as the controls

Supplementary Figure 1

Please see above for questions about the constructs (panel a is not enough information) and please show the lane with all markers for panel c and comment on paxillin degradation

Supplementary Figures 3d and 4 might be clearer in color (thanks for providing these in the rebuttal

and I do still prefer these) but I understand it is not their preference, what is the stain? Please say so in the Figure legend

Supplementary Figure 5

Please provide details on IpaA construct

Please show the lanes with all markers as well as the IpaA alone and (meta)vinculin alone controls, do we know the vinculins do not pellet in the absence of actin?

Supplementary Figure 6b-c

First cartoon, I assume the green is vinculin, the yellow YFP, and the green the tail domain?

Second-fourth cartoons, what is the black and red?

Supplementary Figure 7

Same questions about the constructs, please see above, and here also for panel d, what is the linker between talin and YFP and how do we know it does not open talin, PDB entry 6r9t could suggest that a proper linker might not affect its structure so it is important to know the construct information; I do not see what the red is meant to show (still panel d and actually for panel b, what does "integrated separately" mean?); PDB entry 6r9t suggests that an insert at 450 might be ok so please provide these details (are any additional residues inserted, what is the linker?) (still panel d) as well as for the blue TagBFP (linker etc.)

Supplementary Figure 8

Same questions about the constructs, please see above

Supplementary Figure 9

Please show the entire blot (also not shown in the source data especially for tubulin) including the lanes with the markers as well as the controls

Supplementary Figure 11

How was integrin and catenin visualized?

Supplementary Figure 12

Please show the entire blot (also not shown in the source data) including the lanes with the markers and please comment on degradation for integrin (only shown in the source data file (but all the full blots should be in the supplementary Figures instead of the Excel spreadsheet please and please provide information on antibodies in the legends

With regards to the response of the choice of HL-1 cells, the authors confirm that these cells express vinculin endogenously so wouldn't that be difficult the to deconvolute the effects of the forced expression of their constructs?

The overuse of acronyms has been mitigated to some extend

My comments on the I997A mutant remain; with all respect, we have much more knowledge today than Thievensen and colleagues had 7 years ago and their actin co-sedimentation (Figure 3B) had a lot of degradation for the mutant

Response to Reviewer #3

I am very happy about the thorough revision and additional controls. They have improved the quality of the manuscript significantly and I therefore recommend publication.

Response: We thank the reviewer once again for evaluating our manuscript.

Response to Reviewer #4

I find the apparent addition of the construct information to the Figure legends insufficient; their explanation as to why not describe these in the main manuscript, i.e. the methods section in detail also, is unclear as it would be in everyone's interest to follow the experiments easily:

Response: We thank the reviewer for evaluating our revised manuscript. We agree, it is in everyone's interest that experiments can be easily understood and we therefore tried to realize as many of the reviewer's suggestions as possible. In the second revised version of the manuscript, we include additional sentences into the text, provide an additional citation, and include more information into the source data.

Please note, however, that we could not comply with the reviewer's recommendation to include methodological details in the figure legends, as this appears to be in conflict with the formatting guide of *Nature Communications*, stating: "*Figure legends should be <350 words each. They should begin with a brief title sentence for the whole figure and continue with a short statement of what is depicted in the figure, not the results (or data) of the experiment or the methods used.*"

(https://media.nature.com/full/nature-assets/ncomms/pdf/submission_guide_ncomms.pdf).

In these particular cases, we further specified the experimental details in the methods section. Together with the numerous schematic drawings of the constructs in Supplementary Fig. 1, Supplementary Fig. 3 and Supplementary Fig. 7, this will certainly enable the reader to understand how constructs were generated and experiments were conducted. We sincerely hope that the additional modifications will now allow the reviewer to agree with the recommendations of the other two referees and support the publication of the study.

Figure legend 1a-1c, 1e

The tag is suggested to be on the N-terminus from the methods but on the C-terminus in Supplementary Figure 1a;

Response: We have checked the manuscript carefully and believe that this statement by the reviewer is incorrect. Neither figure legends nor methods section state that tags were fused to the N-terminus. They were added C-terminally. To make this even more clear, we modified the sentence in lines 282–284 that now reads: "*Single fluorophore fusion proteins were generated by incorporating YPet, mCherry, venus(A206K) or TagBFP cDNA, internally or C-terminally, as shown schematically in Supplementary Fig. 1, 3, and 7.*" We hope this clarifies the misunderstanding.

what about the linker region and how is it ensured that the tag does not hinder or force (meta)vinculin opening to its active conformer? This is perhaps answered for the TS constructs in Supplementary Figure 5

Response: Yes, Supplementary Fig. 5 shows that the insertion of the TS module into the linker region does not lead to a constitutive activation of (meta)vinculin. Please note that reviewer #3 raised the same

concern and was satisfied with this additional experiment, shown in Supplementary Fig. 5. Please also note that the rescue of spreading in vinc (-/-) cells by the (meta)vinculin C-terminal fusion constructs (V-V and M-V; Fig. 1b) and the (meta)vinculin tension sensors (V-TS and M-TS; Fig. 2b) is further evidence that the function of vinculin is not compromised by C-terminal tagging or the TS module insertion.

In addition, we would like to emphasize that our original work (Grashoff et al, Nature, 2010) evaluated vinculin function after TS module insertion, for example by analyzing FA turnover rates using FRAP experiments or comparing the vinculin tension sensor with a vinculin conformation sensor. These experiments show that the insertion of the TS module at aa 883 does not significantly affect vinculin conformation; in fact, the data demonstrate that vinculin activation and force transduction are separate events. Furthermore, please note that vinculin tension sensors with an insertion at aa 883 have been independently used by many other research groups, for example the Kumar lab (Chang and Kumar, JCS, 2013), the Hoffman lab (Rothenberg et al, Biophys J, 2018; La Croix et al, Elife, 2018), and the Yap lab (Leerberg et al, Curr Biol, 2014). None of these groups have reported any vinculin impairment. Thus, even if the TS module insertion affects vinculin function, this appears to be a negligible effect with regard to the here described effects.

Figure legend 1d

Same comment for talin constructs as for vinculin in 1a-1c, 1e

Response: We are unsure to which of the above comments the reviewer refers to: N-terminal tagging of constructs, or concerns about talin functionality upon tagging or TS-insertion.

Regarding the tagging of talin-1, please note that the talin construct in panel 1d (T1-B-HA) is depicted and described in Supplementary Fig. 1: “*Talin-1 (grey) C-terminally-tagged with TagBFP (blue) followed by an HA-tag (black)*” (lines 718–719).

Concerning talin functionality after tagging, we note that C-terminal fusions and talin tension sensors have been evaluated in detail by us (Austen et al, Nat Cell Biol, 2015) and others (Kumar et al, JCB, 2016) and were found to recapitulate talin function, as shown by the rescue of cell adhesion, cell spreading, FAK activation and highly similar FA turnover rates in fluorescence recovery after photobleaching (FRAP) experiments. Please also note that genetically modified *Drosophila melanogaster* flies homozygously expressing talin tension sensors or C-terminal talin fusion constructs are all completely normal (Lemke et al, Plos Biology, 2019), which is very strong evidence that the here used strategies of talin tagging do not abrogate talin function.

Figure legend 1f-1g

Please show all markers (the lane is missing with all, not just the nearest to the protein of interest, and what standard is at 151 kDa for example, sorry for not explaining better before) as well as the controls

Response: Please note that we indicate the expected molecular weights of the constructs in the figure. This is now clearly stated in the figure legend. In our lab, we use proteins markers that do not show a

luminescent signal in the western blot. Instead, the prestained marker bands are traced with a pencil on the membrane and recorded in a separate image. These protein marker bands are now all shown in the source data. For clarification, the figure legend includes the following sentence: *“The kDA values indicate the expected molecular weight of the respective protein. Uncropped immunoblots including protein markers are shown in the source data.”*

Regarding controls, we note that all data necessary to conclude that metavinculin shows stronger association with talin are presented: The immunoblots (IBs) indicate that equal amounts of prey protein were present in the lysate; the immunoprecipitation samples IP: venus (Fig. 1f) and IP: talin (Fig. 1g) demonstrate that equal amounts of bait were pulled down; the IP: talin (Fig. 1f) and IP: venus (Fig. 1g) lanes show the increased association of talin and metavinculin.

Figure legend 2

Please describe the constructs in full

Response: Figure 2 shows data from twenty different constructs, which are described in the main text, the methods section, Supplementary Fig. 3, and Supplementary Fig. 7. We do not believe that it would be helpful to repeat this information in the figure legend, instead of describing the presented data.

Please note that this suggestion is also in conflict with the official formatting guide of *Nature Communications* (see above), and we hope the reviewer will understand that we wish to follow these guidelines.

The insertion at residue 883, thank you for now mentioning this, might probably affect its interaction with the helix bundle

Response: As described above, all our experiments (proper subcellular localization, rescue of spreading defect, modulation by IpaA in the actin co-sedimentation assay) indicate that the function of vinculin is not significantly altered by the insertion of the TS module. As indicated above, please also note that our original work (Grashoff et al, Nature, 2010) evaluated vinculin function after TS module insertion, for example by analyzing FA turnover rates using FRAP experiments. These experiments also demonstrated that the insertion of the TS module at aa 883 does not significantly affect vinculin function. As indicated above, note that vinculin tension sensors with an insertion at aa 883 have been independently used by many other research groups, including the Kumar lab (Chang and Kumar, JCS, 2013), the Hoffman lab (Rothenberg et al, Biophys J, 2018; La Croix et al, Elife, 2018), and the Yap lab (Leerberg et al, Curr Biol, 2014). None of these groups have reported any vinculin impairment. If vinculin function is impaired, it appears to be a small and negligible effect.

I cannot find the description for V-TS or M-TS (panels a and c in Supplementary Figure 3 do not have enough details and the reference (or information available on Addgene) does not replace a one sentence description of the constructs;

Response: Please note that, in addition to Supplementary Fig. 3, V-TS and M-TS constructs are described in the main text (lines 78–82), and the (now modified) methods section (lines 278–281).

“lacking the C-terminal actin binding domain” is which residues please? “In parallel, we generated control constructs to determine the fluorescence lifetime of the donor fluorophore as well as the FRET efficiency of the no force control (Con TS), which comprises the vinculin head domain (aa 1–883) and a tension sensor (TS) module but lacks the vinculin tail domain (Supplementary Fig. 3a c).” meaning 884-1061 were deleted?)

Response: Please note that the cited sentence fully describes how the construct was assembled. The word “comprise” means “consist of”, which in the context of the cited sentence literally means that “aa 884-1066” are not included. To be even more clear, we modified the relevant sentence in the methods section and line 281-282 now reads: “No force control constructs were terminated directly after the TS modules, therefore lacking aa 884–1066(1134).”

or FL-based tension sensors

Response: Please note that these constructs are described in the main text (lines 78–82), the modified methods section (lines 278–281), and Supplementary Fig. 3.

or the two F7-TS constructs,

Response: The F7-TS constructs are described in the main text (lines 115–117), and the methods section (lines 284-286).

or Ts-TS or the two B constructs (panel a in Supplementary Figure 1 does not have enough information) etc

Response: The T2-TS, V-B and M-B constructs are described in the main text (lines 125–126), the modified methods section (lines 282–284, 289–291), and Supplementary Figure 7.

Regarding additional information about constructs in figure legends, we again point out that this suggestion is not in line with the formatting guide of *Nature Communications* (see above). Please understand that we wish to follow these guidelines.

Figure 3 legend

Same as for Figure 2 and also not clear how the Ypet fusion is constructed

Response: Please note that Figure 3 does not show a YPet fusion. Instead, images represent YPet signal from vinculin (V-TS) and metavinculin (M-TS) tension sensors, and the no-force control (Con-TS), as is clearly stated in the figure legend.

All three constructs are fully described in the main text (lines 78–85), the modified methods section (lines 278–282), and Supplementary Fig. 3.

The suggestion to repeat this information in this figure legend is in conflict with the formatting guide of *Nature Communications* (see above). We hope the reviewer will understand that we wish to follow these guidelines.

Figure 4

Please show the entire blot (some, for example tubulin, still seem cropped in the source data) including the lanes with the markers as well as the controls

Response: This blot was cut horizontally to allow detection of vinculin (upper half) and tubulin (lower half) without stripping the western blot membrane. The blots were re-assembled in the source data accordingly. Please note that the full-length blot contains also a Tln2-antibody staining that is not shown in Fig. 4. Furthermore, marker bands are now included in the source data and we added the following sentence to the figure legend: *“The kDA values indicate the expected molecular weight of the respective protein. Uncropped immunoblots including protein markers are shown in the source data.”*

Supplementary Figure 1

Please see above for questions about the constructs (panel a is not enough information) and please show the lane with all markers for panel c and comment on paxillin degradation

Response: We hope that we have addressed all the questions regarding used constructs in the responses above. Again, we hope that the reviewer will respect our wish to comply with the *Nature Communications* formatting guide.

Marker bands are now included in the source data and we added the following sentence into the figure legend: *“The kDA values indicate the expected molecular weight of the respective protein. Uncropped immunoblots including protein markers are shown in the source data.”*

Concerning the paxillin blot, we note that the paxillin double band at 68 kDa (shown in Supplementary Fig. 1c) is a result of phosphorylation, and consistent with previously published data (for example: Pasapera et al, JCB, 2010: “Myosin II activity regulates vinculin recruitment to focal adhesions through FAK-mediated paxillin phosphorylation”, Fig. 5) and the information found in the antibody data sheet (<https://www.bdbiosciences.com/ds/pm/tds/610051.pdf>).

The additional band below (seen in the source data) appears to be unspecific and is also visible in the data sheet picture (please see on the right).

As this unspecific band occurs in all lysates, our conclusion that the expression of vinculin or metavinculin does not alter the expression of the here tested FA proteins remains valid.

Supplementary Figures 3d and 4 might be clearer in color (thanks for providing these in the rebuttal and I do still prefer these) but I understand it is not their preference, what is the stain? Please say so in the Figure legend

Response: We thank the reviewer for respecting our preference. Even though it is not entirely consistent with the *Nature Communications* formatting guide, we now explain in more detail how the stainings were performed in the modified sentence: “FAs were visualized with a paxillin antibody in cells spreading for 4 h on FN coated glass slides; f-actin was labeled with phalloidin.”

Supplementary Figure 5

Please provide details on IpaA construct.

Response: The IpaA construct was originally described by DeMali et al (“IpaA targets beta1 integrins and rho to promote actin cytoskeleton rearrangements necessary for Shigella entry” JBC, 2006) and purified according to the protocol in the referenced manuscript (26). This is now specified in the methods section of the manuscript (lines 313-314). We also provide the reference for the DeMali study, which is now listed as reference 55.

Please show the lanes with all markers as well as the IpaA alone and (meta)vinculin alone controls, do we know the vinculins do not pellet in the absence of actin?

Response: Protein markers are now shown in the source data. Please note that in the figure we indicate the expected molecular weights of the constructs. The figure legend now reads: “The kDa values indicate the expected molecular weight of the indicated protein. Uncropped immunoblots including protein markers are shown in the source data”.

The “IpaA alone” and “(meta)vinculin alone” experiments were not performed because the sole purpose of the experiment was to test whether the addition of IpaA can shift V-TS and M-TS from the soluble into the actin pellet fraction. As shown in Supplementary Fig. 5, this is the case; please also note that reviewer #3 was satisfied with this experiment.

Supplementary Figure 6b-c

*First cartoon, I assume the green is vinculin, the yellow YFP, and the green the tail domain?
Second-fourth cartoons, what is the black and red?*

Response: Please note that these schematics were introduced in Supplementary Fig. 3 and are reused in this figure. Even though we find this to be intuitive, we have added another sentence into the figure legend of Supplementary Fig 6 to point this out: *“The schematic drawings of protein constructs in (b) and (c) are described in detail in Supplementary Fig. 3a, c.”*

We also added the description of the red line and black dots in the plots and now state: *“To analyze FRET efficiencies, the fluorescence lifetime of donor-only-lifetime control (V-Y), no-force control (Con-TS) and tension sensors (V-TS and M-TS) are determined by fitting a mono-exponential decay function (red line) to the photon count time trace (black dots) of each masked cell”.*

Supplementary Figure 7

Same questions about the constructs, please see above, and here also for panel d, what is the linker between talin and YFP and how do we know it does not open talin, PDB entry 6r9t could suggest that a proper linker might not affect its structure so it is important to know the construct information; I do not see what the red is meant to show (still panel d and actually for panel b, what does “integrated separately” mean?); PDB entry 6r9t suggests that an insert at 450 might be ok so please provide these details (are any additional residues inserted, what is the linker?) (still panel d) as well as for the blue TagBFP (linker etc.)

Response: We hope that we have addressed all the questions regarding used constructs in the responses above. Again, we hope that the reviewer will respect our wish to comply with the *Nature Communications* formatting guide.

Please note that, in addition to Supplementary Fig. 7, all talin-2 constructs are described in the main text (lines 125–128) and the modified method section (lines 289–292), where we now specify the linkers between talin-2 and the fluorophores, as requested.

Regarding talin functionality, please note our previously published study demonstrating that the insertion of the TS module into talin-2 or the C-terminal tagging of talin-2 do not compromise talin function (Austen et al, Nat Cell Bio, 2015). All constructs can rescue the integrin activation defect of talin-deficient cells, induce cell spreading, FA formation and FA maturation.

The red color indicates the acceptor mCherry as described in legends of Supplementary Fig. 1, Supplementary Fig. 3a and Supplementary Fig. 7. Note that we slightly changed the figure legend of

Supplementary Fig. 3 to make this even more clear. The sentence in line 740-742 now reads: “The TS module, comprising donor (YPet) and acceptor (mCherry) fluorophores that are connected by a mechanosensitive linker, is integrated between head and tail region of vinculin.”

Moreover, we have modified the sentence at the end of this figure legend (lines 810-812), which now states: “Elements in all schemes: YPet and venus donor fluorophores (yellow), mCherry acceptor fluorophore (red), vinculin (green), metavincludin insert (purple), mechanosensitive linker (black).”

“Integrated separately” means that either the donor (venus) or the acceptor (mCherry) were integrated, as indicated schematically in Supplementary Fig. 7b.

Supplementary Figure 8

Same questions about the constructs, please see above

Response: Please note that all constructs are fully described in the main text (lines 78–85), the modified methods section (lines 278–282), and Supplementary Fig. 3. Again, we hope that the reviewer will respect our wish to comply with the *Nature Communications* formatting guide and not repeat details on experimental methods in the figure legend.

Supplementary Figure 9

Please show the entire blot (also not shown in the source data especially for tubulin) including the lanes with the markers as well as the controls

Response: Please note that this blot was cut horizontally, to allow detection of vinculin (upper half) and tubulin (lower half) without stripping. Full blots and markers are now shown in the source data and the figure legends include the following sentence: “*The kDA values indicate the expected molecular weight of the respective protein. Uncropped immunoblots including protein markers are shown in the source data*”. Please note that the full blot contains samples that are not shown in the Supplementary Fig. 9.

Supplementary Figure 11

How was integrin and catenin visualized?

Response: We apologize if we do not understand this question, but all information is available in the manuscript. To visualize integrin and catenin, tissue samples were processed as described in the methods section “Immunostaining and immunohistochemistry”, using the antibodies and dilutions listed in the methods section “Antibodies and Reagents”. Images were acquired using a Zeiss LSM780 microscope, as described in the methods section “Immunostaining and immunohistochemistry”.

Supplementary Figure 12

Please show the entire blot (also not shown in the source data) including the lanes with the markers and please comment on degradation for integrin (only shown in the source data file (but all the full blots should be in the supplementary Figures instead of the Excel spreadsheet please and please provide information on antibodies in the legends

Response: Please note that western blotting of tissue lysates is a quite challenging task. Tissue samples often give high background signals and antibodies may detect more unspecific bands, as compared to samples from cellular lysates. This is probably caused by the numerous extracellular matrix components present at high concentration in the tissue lysates. We therefore repeated western blotting experiments and horizontally cut membranes to be able to establish ideal conditions for protein detection without stripping the western blot membrane. We have now re-assembled all these blots in the source data of Supplementary Fig. 12 and black boxes indicate which signals were assembled in the Supplementary Fig. 12. During the assembly we noted that a β -catenin blot (from six months old mice) was inverted vertically. We apologize for this mistake but note that this does not change our scientific conclusions. This is now also corrected in the Supplementary Fig. 12 and in the source data. In addition, markers are now shown in the source data file and the figure legends include the following sentence: “*The kDA values indicate the expected molecular weight of the respective protein. Uncropped immunoblots including protein markers are shown in the source data*”.

The reason for the degradation band observed in two samples on the integrin blot is not entirely clear. We note, however, that tissue samples seem especially prone to protein fragmentation, presumably because of high protease activity in certain tissues and the extended time required to prepare tissue lysates. Therefore, the most likely explanation is that the observed degradation is an artefact of the sample preparation, which is consistent with the fact that a) the degradation is not observed in all samples, b) mice do not show a phenotype, and c) such degradation bands were not observed in the samples from 13-month-old mice. Therefore, we would prefer not to feature this (most probably unspecific) band in the Supplementary Fig. 12, because we are worried that – without extensive explanation – it might cause an undue confusion and distract from the actual data. However, we now write in lines 862-864 of the figure legend that a protein degradation was observed and can be viewed in the source data.

Regarding the issue of listing antibodies in the figure legends, please understand that we refer to the formatting guide of *Nature Communications* stating that figure legends should not describe experimental details. However, we firmly believe that any reader who is interested in repeating the experiments can find all information on antibodies in the methods section including manufacturer, ordering number and the used dilution of antibodies.

Regarding the suggestion to show all blots in the Supplementary Figures, we would like to follow the policies of *Nature Communications* that can be found at: <https://www.nature.com/ncomms/journal-policies/editorial-publishing-policies>. They state: “*For relevant manuscripts, we may request a source data file in Microsoft Excel format or a zipped folder. The source data file should, as a minimum, contain the*

raw data underlying any graphs and charts, and uncropped versions of any gels or blots presented in the figures." We hope the reviewer will understand that we wish to follow these official guidelines.

With regards to the response of the choice of HL-1 cells, the authors confirm that these cells express vinculin endogenously so wouldn't that be difficult the to deconvolute the effects of the forced expression of their constructs?

Response: We do not attempt to deconvolute effects of protein expression, and we also do not think that this is necessary for making the conclusion drawn in this manuscript. Our experiments simply show that the expression and analysis of (meta)vinculin tension sensor in HL-1 cells results in very similar FRET efficiency differences observed in fibroblasts. We conclude that effects seem to be conserved in different cell types.

Please note that vinculin tension sensors have been used to study forces in the presence of endogenous proteins before. The cell migration assays in Grashoff et al (Nature, 2010, Fig. 3), for instance, were performed in bovine aortic endothelial cells (BAECs) which also express endogenous vinculin. Another, quite impressive example is a recent study by Tao et al (Nature Communications, 2019), who generated transgenic mice to study vinculin tension *in vivo*. Also in these experiments, the vinculin tension sensor is expressed in addition to the endogenous protein showing that a deconvolution is not required to learn about processes of molecular force transduction using the vinculin tension sensor system.

The overuse of acronyms has been mitigated to some extent

Response: Thank you, we are glad that the reviewer acknowledges our efforts. The fact that this study uses a great number of rather complex expression constructs makes the use of acronyms unavoidable.

My comments on the I997A mutant remain; with all respect, we have much more knowledge today than Thievensen and colleagues had 7 years ago and their actin co-sedimentation (Figure 3B) had a lot of degradation for the mutant

Response: We hope the reviewer will agree that our experiments in Fig. 2c (and Supplementary Fig. 7a) clearly demonstrate that the insertion of the I997A mutation into vinculin and the insertion of the I1065A mutation into metavinculin abrogate FRET efficiency differences in the tension sensor measurements. This is the scientific finding that we report here and this finding remains valid.

In light of the published literature, we find it reasonable to suggest that those effects are related to a reduced association of (meta)vinculin with actin. We are not sure which alternative explanation the reviewer would prefer, but we are confident that scientific progress will clarify how those mutants affect vinculin function.